# EMS: Adaptive Evict-then-Merge Strategy for Head-wise KV Cache Compression Based on Global-Local Importance

## Abstract

As large language models (LLMs) continue to advance, the demand for higher quality and faster processing of long contexts across various applications is growing. KV cache is widely adopted as it stores previously generated key and value tokens, effectively reducing redundant computations during inference. However, as memory overhead becomes a significant concern, efficient compression of KV cache has gained increasing attention. Most existing methods perform compression from two perspectives: identifying important tokens and designing compression strategies. However, these approaches often produce biased distributions of important tokens due to the influence of accumulated attention scores or positional encoding. Furthermore, they overlook the sparsity and redundancy across different heads, which leads to difficulties in preserving the most effective information at the head level. To this end, we propose EMS to overcome these limitations, while achieving better KV cache compression under extreme compression ratios. Specifically, we introduce a Global-Local score that combines accumulated attention scores from both global and local KV tokens to better identify the token importance. For the compression strategy, we design an adaptive and unified Evict-then-Merge framework that accounts for the sparsity and redundancy of KV tokens across different heads. Additionally, we implement the head-wise parallel compression through a zero-class mechanism to enhance efficiency. Extensive experiments demonstrate our SOTA performance even under extreme compression ratios. EMS consistently achieves the lowest perplexity, improves scores by over 1.28 points across four LLMs on LongBench under a 256 cache budget, and preserves 95% retrieval accuracy with a cache budget less than 2% of the context length in the Needle-in-a-Haystack task.

## 1 Introduction

Large language models (LLMs) (Devlin, 2018; Brown et al., 2020; Anil et al., 2023; Dubey et al., 2024; Jiang et al., 2023) have demonstrated remarkable capabilities across various domains, such as question answering (Jiang et al., 2021; Lazaridou et al., 2022), retrieval systems (Ram et al., 2023; Xu et al., 2023), logical reasoning (Wei et al., 2022; Liu et al., 2023a), and code generation (Roziere et al., 2023; Liu et al., 2024a), *etc*. With growing application demands for LLMs, the requirement to manage long sequences (Chen et al., 2024b; Jin et al., 2024; Chen et al., 2023) is also increasing. So far, GPT-4 (Achiam et al., 2023) can process approximately 128K tokens, Gemini-Pro-1.5 (Team et al., 2023) handles about 1M tokens, and Kimi-Chat can process up to 2M tokens. These developments pose significant challenges to the inference efficiency of LLMs. One key acceleration technique is the KV cache, where the key-value (KV) states generated during inference are stored in GPU memory to avoid redundant computation and improve processing efficiency. However, the size of the KV cache grows with the length of the input sequence, which severely limiting the applicability of LLMs (Yuan et al., 2024). Therefore, efficiently compressing the KV cache while preserving essential information has become a critical issue.

Extensive researches have been carried out to address it. Specifically, StreamingLLM (Xiao et al., 2024) first discovered the sink mechanism and achieved infinite output by retaining the initial and local tokens. Subsequent works mainly approached this issue from two perspectives: (*i*) extracting

the most important parts of the generated KV tokens, and (*ii*) designing compression strategies to preserve more information. For example, H2O (Zhang et al., 2023) focuses on globally important tokens, while SnapKV (Li et al., 2024b) concentrates on tokens with higher relevance within a local window size. However, due to cumulative effects of attention weights and positional encoding, the selected tokens exhibit a biased tendency: H2O favors earlier information, while SnapKV leans towards later context. Building upon lightweight token compression methods (Bolya et al., 2022; Yin et al., 2022; Li et al., 2024a), existing approaches are generally categorized into evict-based and merge-based methods. However, a unified and effective approach that combines evict and merge for extreme KV cache compression is still largely unexplored. Therefore, we pose the following question:

***Can we select important KV tokens in a more balanced manner while retaining as much information as possible at a high compression ratio?***

To answer this question, we decouple the compression of KV cache into two stages: selecting important tokens and compressing the KV cache based on those selections. Under this workflow, we propose EMS, a head-wise **E**vict-then-**M**erge **S**trategy based on Global-Local importance, as illustrated in Figure 1. Building on the token importance bias, we dynamically integrate the global- and local-aware importance and construct a more balanced Global-Local score, which serves as the fundamental indicator for efficient compression. Specifically, the score is calculated by aligning the accumulated attention scores of all KV tokens and recent KV tokens. This approach not only ensures a more balanced selection of important KV tokens, but also mitigates biases caused by attention accumulation and positional encoding.

For the compression strategy, the observed differences in sparsity and redundancy across different heads suggest that applying head-wise eviction and merging can potentially achieve a higher compression ratio. Building on this, we propose a unified Evict-then-Merge strategy at the fine-grained head level to improve storage density. In particular, the most irrelevant tokens will first be evicted, leaving only the tokens with higher importance scores for further compression. A subset of the remaining tokens is then selected as class centers based on their higher importance, with less important tokens being merged into these centers. However, as not all tokens are suitable for merging due to low redundancy, those with low similarity to important KV tokens are evicted to minimize output disturbance. To ensure parallel inference during head-wise merging and eviction, we introduce a zero-class center where evicted tokens are merged, treating the eviction process as a special case of merging. Additionally, this step allows for a dynamic merge ratio for each head, ensuring a more adaptive and efficient compression. EMS achieves extreme compression while preserving the capabilities of LLMs, boosting scores by more than 1.28 points on four LLMs in LongBench with a 256 cache budget, and retaining 95% retrieval accuracy using under 2% of the context length in the Needle-in-a-Haystack task. To summarize, our contribution can be generalized as:

(1) **We design a more balanced Global-Local score for important token selection.** Our Global-Local score automatically integrates global and local attention of KV tokens, reducing bias and ensuring more balanced token selection across different tasks.

(2) **We propose a sparsity- and redundancy-driven Evict-then-Merge compression strategy.** Based on the head-wise sparsity and redundancy characteristic of KV tokens, we develop an Evict-then-Merge strategy that maximizes information retention even under low compression ratios.

(3) **We implement an efficient head-wise parallel compression for the KV cache.** We propose a zero-class center where KV tokens with low similarity to important KV tokens are merged. In this way, we treat eviction as merging, achieving input-aware parallel evict and merge ratios, while maintaining a constant budget for each head. This strategy efficiently leverages the head-wise distribution property in a simple yet effective way.

## 2 RELATED WORK

### 2.1 IMPORTANT KV SELECTION

Recent works on KV cache compression have focused on the selection of the important KV tokens to preserve the performance of uncompressed LLMs. StreamingLLM (Xiao et al., 2024) discovered the sink mechanism, where the attention weights for the initial tokens are significantly high

regardless of its meaning. Consequently, it retains the first four tokens along with some local tokens, enabling streaming output with the constrained memory. Following it, ACT (Yu et al., 2024) further demonstrates that certain KV tokens in the middle also exhibit high attention weights. H2O (Zhang et al., 2023) selects important KV pairs based on accumulated attention score, leading to more concentration in the former part of the context. SnapKV (Li et al., 2024b) observed that calculating attention within a local window of the prompt can capture specialized attention features. This approach retains more recent tokens, which are effective for local generation. However, in most scenarios, valuable information is often spread across the contexts and varies from different tasks. Therefore, the selection of important KV pairs should be tailored to the specific input and avoid former or recent preference caused by importance indicator.

## 2.2 KV COMPRESSION

Inspired by various token compression methods (Yin et al., 2022; Bolya et al., 2022; Li et al., 2024a), KV cache compression can also be categorized into eviction and merge strategies. Evict-based methods aim to retain only the important KV pairs under different token importance assumption (Xiao et al., 2024; Zhang et al., 2023; Li et al., 2024b; Adnan et al., 2024; Cai et al., 2024; Feng et al., 2024), while maintain the modeling ability. DCP (Anagnostidis et al., 2023) introduces a lightweight attention block in each layer to dynamically decide which KV pairs to discard based on the input. FastGen (Ge et al., 2023) extends this by finding that different heads focus on different token types. Through the attention profiling in the prompt prefilling stage, it defines distinct compression strategies for each head, which are applied during inference. However, this static approach can be suboptimal in long context scenarios where attention patterns change as decoding proceeds. Besides, compressing each head differently can lead to uneven head budgets, making storage and computation more difficult.

Compared to evict-only methods, merge-based approaches have the potential to retain more information and enhance model performance. CAM (Zhang et al., 2024) analyzes the attention output error caused by token eviction, only merge the evicted value tokens into the remaining ones to reduce performance loss. DMC (Nawrot et al., 2024) dynamically decides whether to merge the current token to the tail of the KV cache in a head-wise manner. However, compression only at the tail may not be the most effective approach. LESS (Dong et al., 2024) introduces a low-rank embedding sidekick with sparse policy, which accumulates the information discarded by the eviction strategy into a fixed-size low-rank cache. Besides, DMC and LESS require additional training for performance gains. It is worth noting that KV cache merging is different from token merging, due to the paired processing of keys and values and the autoregressive characteristic of LLMs. Improper merging methods can lead to significant error accumulation in the decoding stage, resulting in severe performance degradation. Building on these insights, we propose an efficient Evict-then-Merge compression method that effectively addresses these challenges.

## 3 HYBRID TOKEN SELECTION POLICY BASED ON GLOBAL-LOCAL SCORE

### 3.1 PRELIMINARIES

Due to the autoregressive nature of the LLM inference process, the key and value states calculated in previous timesteps are repeatedly used for attention. To avoid computational redundancy, LLMs can store previously computed key and value states for future generation, which is known as KV cache.

In particular, given $n$ prompt tokens, the model first prefills the prompt information to the key and value states, and KV cache is initialized by $C_K^0 = (\boldsymbol{k}_1, \boldsymbol{k}_2, \cdots, \boldsymbol{k}_n)$ and $C_V^0 = (\boldsymbol{v}_1, \boldsymbol{v}_2, \cdots, \boldsymbol{v}_n)$. During the decoding stage, the newly generated tokens will be fed into the model, the corresponding key and value states will be appended to $C_K$ and $C_V$. Taking timestep $t$ as an example, the model computes $\boldsymbol{q}_t$, $\boldsymbol{k}_t$ and $\boldsymbol{v}_t$, and loads the previous key and value states. The KV cache is updated by $C_K^t = [C_K^{t-1}, \boldsymbol{k}_t]$ and $C_V^t = [C_V^{t-1}, \boldsymbol{v}_t]$, and the attention is calculated by:

$$\text{Attention}(\boldsymbol{q}_t, \boldsymbol{K}_t, \boldsymbol{V}_t) = \text{softmax}\left(\frac{\boldsymbol{q}_t \left[C_K^{t-1}, \boldsymbol{k}_t\right]^T}{\sqrt{d}}\right) \left[C_V^{t-1}, \boldsymbol{v}_t\right], \tag{1}$$

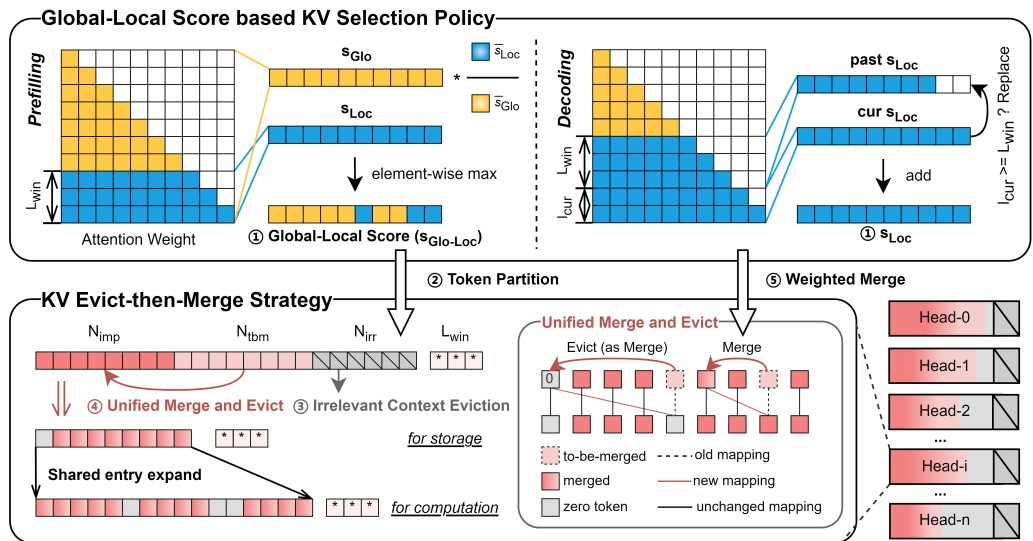

Figure 1: The framework of EMS. The compression of KV cache is decoupled into two parts. For important KV selection policy, a balanced Global-Local score is designed to grasp token importance. For KV compression strategy, the Evict-then-Merge approach first removes irrelevant tokens, then applies a unified head-wise eviction and merging process.

where $\mathrm{softmax}$ is the normalized exponential function and $d$ is the feature dimension. Afterwards, $\boldsymbol{C}_K^t$ and $\boldsymbol{C}_V^t$ are stored and will be loaded for subsequent generation. However, the memory overhead of KV cache grows linearly as LLMs inference progresses, which greatly limits the inference efficiency. Therefore, efficiently compressing the KV cache while preserving essential information is a critical challenge.

## 3.2 TOKEN IMPORTANCE BIAS

Existing methods for assessing the most essential KV tokens exhibit inherent biases. We categorized them into three types that impact compression: local-only bias, local-aware bias, and global-aware bias, as illustrated in Figure 2.

Building on the attention sink (Xiao et al., 2024; Yu et al., 2024), **local-only methods** primarily focus on a few initial sink tokens and the recent ones within a local window. Although they exhibit strong capabilities in local language modeling, their performance drops significantly in tasks requiring global semantic understanding, such as summarization and full-text comprehension. To select more informative KV tokens, **global-aware methods** (Zhang et al., 2023; Wang et al., 2021) employ the global accumulated attention score to select important tokens, which can be calculated by $s_{Glo} = \sum_{i=1}^{N} \boldsymbol{A}_{i,:}$, where $N$ is the context length, and $\boldsymbol{A} \in \mathbb{R}^{N \times N}$ is the attention weight. However, the cumulative effect of causal attention weight skews the importance distribution towards the earlier part, resulting in a biased selection of important KV tokens and leading to suboptimal outcomes. In contrast, **local-aware methods** Li et al. (2024b); Liu et al. (2023b) utilize local tokens as anchors to identify and retain tokens across the entire KV cache. The accumulated at-

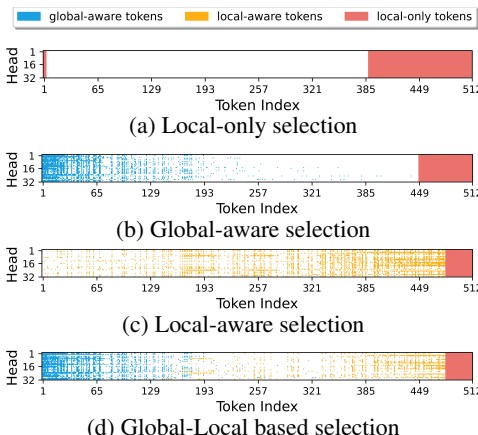

Figure 2: Token selection patterns. We visualize the selection patterns of previous KV cache methods and our method. The sample is taken from the *gov_report* (Huang et al., 2021) dataset, showing Top-128 selected tokens out of a total of 512 tokens.

tention score of local tokens is used to measure the importance, which can be formualted by $s_{Loc} = \sum_{i=N-L_{win}}^{N} A_{i,:}$, where $L_{win}$ is the window size. However, the positional encoding causes nearby tokens to exhibit higher correlations, leading to a bias towards retaining more recent tokens.

### 3.3 Global-Local Score For Non-local Tokens

To eliminate the influence of cumulative effects and positional encoding, we propose a Global-Local score to leverage both global and local information, which is shown in Figure 1. Specifically, we convert $\mathbf{s}_{Glo}$ and $\mathbf{s}_{Loc}$ to similar magnitudes by mean-alignment, followed by the element-wise max to obtain the new score:

$$s_{Glo-Loc} = \max\left(s_{Glo} \times \frac{\sum s_{Loc}/N}{\sum s_{Glo}/N}, s_{Loc}\right), \tag{2}$$

where $\max$ is an element-wise function, $N$ is the context length.

During the prefilling stage, we calculate the accumulated attention scores of $A$ from global and local scope to get $s_{Glo}$ and $s_{Loc}$. These two critical vectors capture the importance distribution of the prompt and are stored for subsequent generation, while the temporary attention weight matrix is deallocated. During the decoding stage, $s_{Glo}$ is update by accumulating the attention from new tokens. And the local tokens in the window will dynamically change. To avoid potential overhead, we set a past score $s_{Loc}^{past}$ to record attention information in the last window, and a current score $s_{Loc}^{cur}$ to accumulate attention from the new queries in the current window. Once the number of tokens in the current window reaches $L_{win}$, $s_{Loc}^{cur}$ will be assigned to $s_{Loc}^{past}$ and reset to zero. The final local score $s_{Loc} = s_{Loc}^{past} + s_{Loc}^{cur}$. So in the decoding stage, the window size used to select local-aware tokens is actually $L_{win} \sim 2L_{win} - 1$.

Under the Global-Local score, the selection of the important KV tokens becomes more balanced, as shown in Figure 2d. In addition, we also conduct a statistic on the contribution of global and local importance to the Global-Local score in Appendix E.

## 4 Head-wise Evict-then-Merge Strategy for KV Cache

### 4.1 Head-wise Sparsity and Redundancy

Previous works (Zhang et al., 2023; Liu et al., 2023b; Ge et al., 2023) have uncovered the sparsity of the KV cache. We further observed that the redundancy within the KV cache is also significant. Taking a holistic view of both sparsity and redundancy presents an opportunity to achieve higher compression ratios.

**Sparsity**. Only a small portion of important tokens contributes to a significant percentage of the $\mathbf{s}_{Glo-Loc}$ score. The sparsity rate of each attention head is determined by the minimum percentage of important tokens retained per head that can achieve over $\zeta$ of the total score:

$$p_m = 1 - \frac{1}{N} \cdot \underset{N_k \in [1,N]}{\arg\min}\left\{N_k \mid \sum_{i=1}^{N_k} s_i \geq \zeta\right\}, \text{ where } s = \text{sort}(s_{Glo-Loc}), \tag{3}$$

where $p_m$ is the sparsity rate, $m$ is the head index, $\text{sort}(\cdot)$ reorders the vector in descending way.

**Redundancy**. Cosine similarity is a reliable metric for cross position token redundancy. Most existing work considers the characteristic of either keys or values individually, applying the same compression strategy to the other, rather than jointly considering KV pairs. Considering that the superimposed positional encoding can attenuate the similarity, we analyze the similarity of raw KV tokens. As shown in Figure 3a, the similarity among key tokens is significantly higher, whereas the similarity among value tokens is relatively lower. If we consider only the key's characteristic, the substantial merging error of value tokens can severely affect generation, ultimately diminishing the compression potential of redundancy. Therefore, we jointly consider key similarity and value similarity, and the redundancy is defined as:

$$R_{i,j} = \cos(k_i, k_j) \cdot \cos(v_i, v_j) = \frac{k_i k_j^T}{\|k_i\|\|k_j^T\|} \cdot \frac{v_i v_j^T}{\|v_i\|\|v_j\|}. \tag{4}$$

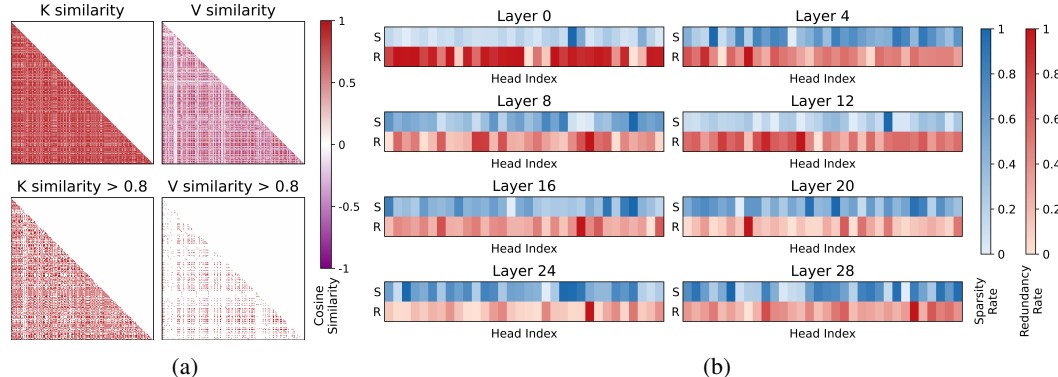

(a)                                                                 (b)

Figure 3: Observations on sparsity and redundancy. The parameters $\zeta$ and $\tau$ are set to 0.95 and 0.6 here. (a) The distribution difference between key and value similarities. The top two figures depict the raw similarity, while the bottom two showcase the masked KV similarities with a threshold of 0.8. Key similarity is much more salient than value similarity. (b) The head-wise sparsity and redundancy. The blue bars represent the sparsity of each head, while the red bars denote the redundancy. Both sparsity and redundancy vary across different heads and layers.

A token is considered redundant if it can find another token in the existing KV cache with redundancy exceeding the predefined threshold $\tau$. The redundancy rate $\boldsymbol{r}_m$ of head $m$ is given by:

$$\boldsymbol{r}_m = \frac{1}{N} \cdot \sum_{k=1}^{N} \left[ \max(\boldsymbol{R}_{k,1:k}) \geq \tau \right].\tag{5}$$

The head-level sparsity and redundancy rates for different heads are shown in Figure 3b.

**Head-wise Sparsity and Redundancy**. From an overall perspective, the earlier layers exhibit more redundancy, while the later layers display greater sparsity. Within each layer, different heads also demonstrate varying degrees of sparsity and redundancy. Consequently, dynamically evicting and merging tokens based on the characteristic of each head can lead to a higher compression ratio. However, the challenge arises from the fact that all heads within a layer are stored and computed in parallel, making it difficult to manage fine-grained evict and merge decisions for each head. In the following section, we propose an efficient, parallelizable, and head-wise solution.

### 4.2 Adaptive Evict-then-Merge Strategy

**Token Partition**. The Global-Local score can effectively measure the importance of tokens. To avoid extract fragmented information, we apply a mean pooling function to it. Based on the ranking of $\boldsymbol{s}_{Glo-Loc}$, KV tokens can be divided into three sets: $N_{irr}$ irrelevant tokens $(\boldsymbol{K}_{irr}, \boldsymbol{V}_{irr})$, $N_{tbm}$ to-be-merged (TBM) tokens $(\boldsymbol{K}_{tbm}, \boldsymbol{V}_{tbm})$, and $N_{imp}$ most important tokens $(\boldsymbol{K}_{imp}, \boldsymbol{V}_{imp})$, except that $L_{win}$ local tokens $(\boldsymbol{K}_{loc}, \boldsymbol{V}_{loc})$ are always kept to preserve the local modeling capability of LLMs, which is crucial for the performance across various tasks. The most irrelevant tokens are first evicted to streamline the process and minimize the impact of irrelevant context (Shi et al., 2023). Then, the most important tokens will serve as class centers based on the cache budget $N_{budget} = N_{imp} + L_{win}$, while $N_{tbm}$ sub-important tokens will be merged into them to preserve as much valuable information as possible. The merge magnification factor $\gamma = (N_{budget} + N_{tbm})/N_{budget}$ indicates the extent to which tokens are merged relative to the cache budget.

**Unified Evict-then-Merge Strategy.** By calculating the redundancy between TBM tokens and class center tokens, the merge destinations for TBM tokens can be identified:

$$\boldsymbol{d}_i = \arg\max_d(\boldsymbol{R}_{i,d}),\tag{6}$$

where $\boldsymbol{d}_i$ is the merge destination for $i$-th TBM token.

To minimize output perturbation, only the tokens with redundancy $\boldsymbol{R}_{i,:d_i}$ exceeding a threshold $\tau$ will be merged, while others being evicted. Inspired by the virtual neighbor technique (Li et al., 2023b; Liu et al., 2024c), we introduced a zero class center to unify the merge and eviction operations, and the evicted tokens are merged into it. Besides, we find that some critical tokens are

highly sensitive to merging, preserving these most significant tokens intact yields greater benefits than sharing them through merging.

Moreover, the varying sparsity and redundancy of each head results in different eviction and merge ratios, as well as varying cache sizes. We choose to allocate an equal budget to each head, using a limited number of entries to store class centers. Based on this, the merged tokens share the same entry in the KV cache, reducing the overall size of the stored KV cache. During the computation of each layer, a smaller cache is loaded and the context length is enlarged through shared entry expanding. More implementation details will be elaborated in Appendix A.

**Attention Score Weighted Merge.** To ensure that more relevant tokens dominate the merged result, attention scores are integrated into the merge process. Given two tokens $(\boldsymbol{k}_i, \boldsymbol{v}_i)$ and $(\boldsymbol{k}_j, \boldsymbol{v}_j)$ to be merged, we apply a weighted merge based on their local importance $\boldsymbol{s}_{Loc}$. As mentioned in Section 3.2, attention scores are crucial for identifying token importance. Therefore, we preserve the norm of key tokens, a scalar for each token, to maintain the accuracy of the attention weights. The merged tokens are given by:

$$\hat{\boldsymbol{k}}_{merged} = \frac{\boldsymbol{w}_i \hat{\boldsymbol{k}}_i + \boldsymbol{w}_j \hat{\boldsymbol{k}}_j}{\boldsymbol{w}_i + \boldsymbol{w}_j}, \ \boldsymbol{v}_{merged} = \frac{\boldsymbol{w}_i \boldsymbol{v}_i + \boldsymbol{w}_j \boldsymbol{v}_j}{\boldsymbol{w}_i + \boldsymbol{w}_j}, \tag{7}$$

where $\boldsymbol{w}_i$ is the local importance of $i$-th token, $\hat{\boldsymbol{k}}_i = \frac{\boldsymbol{k}_i}{\|\hat{\boldsymbol{k}}_i\|}$ and $\hat{\boldsymbol{k}}_j = \frac{\boldsymbol{k}_j}{\|\hat{\boldsymbol{k}}_j\|}$ are normalized key tokens. $(\|\boldsymbol{k}_i\|\hat{\boldsymbol{k}}_{merged}, \boldsymbol{v}_{merged})$ and $(\|\boldsymbol{k}_j\|\hat{\boldsymbol{k}}_{merged}, \boldsymbol{v}_{merged})$ are actually used for computation.

## 5 EXPERIMENT

### 5.1 SETTINGS

In this paper, we use models from both the Llama family (Llama-2-7B (Touvron et al., 2023), Llama-3-8B (Dubey et al., 2024)) and Mistral series (Jiang et al., 2023), including not only original pretrained model, but also instruction-tuned versions such as LongChat-7B-v1.5-32k (Li et al., 2023a) and Mistral-7B-Instruct-v0.2, which can handle up to 32k context length. For comparison, we benchmark our method against StreamingLLM, CAM, H2O, and SnapKV. All these methods compress the KV cache during both the prefilling and decoding stages, except for SnapKV, which compresses only the prompt after the prefilling stage. All experiments can be conducted on a single NVIDIA A100 GPU with 40GB of memory, except for the fully cached model. For common parameters, both EMS and SnapKV use $L_{win} = 32$, kernel_size = 7 for perplexity and LongBench, and $L_{win} = 16$, kernel_size = 7 for Needle-in-a-Haystack task.

### 5.2 PERFORMANCE EVALUATION ON LONGBENCH

LongBench (Bai et al., 2023) is a comprehensive benchmark designed to assess long context modeling abilities across 6 categories of tasks. To enhance performance on question-answering tasks, we employ instruction-tuned models such as LLaMA-2-7B-Chat, LLaMA-3-8B-Instruct, LongChat-7B-v1.5-32k, and Mistral-7B-Instruct-v0.2. To demonstrate the effectiveness of EMS under extreme KV cache compression, we enforce a strict compression budget of 256, with $\tau$ and $\gamma$ set to approximately 0.6 and 4. To adapt the calculation of global attention score on a single GPU for long-context models, we compute scores by retaining half of the tokens from both the start and the end, following observations from "Lost in the Middle" (Liu et al., 2024b), where LLMs tend to have a better grasp of the information at the beginning or end of the input context.

The results are shown in Table 1. We present two versions of our method: one without position information to merge more aggressively and capture global information, and one with position information for tasks requiring precise token localization. This distinction is evident in their task performance. As shown in the table, the baseline methods show varying degrees of performance across different tasks, and the optimal compression strategy differs between models. In contrast, our method integrates the strengths of the baselines, achieving the state-of-the-art performance on nearly all tasks, with improvements of 1.28, 1.79, 17.64, and 1.28 across the four different LLMs. Notably, on LongChat-7B-v1.5-32k, while the performance of other compression methods breaks down, our method consistently maintains high performance, demonstrating its robustness under extreme compression ratios and long-context conditions.

Table 1: Performance evaluation on LongBench across four LLMs. All methods are tested under cache budget 256, except for SnapKV, which increases this budget at the decoding stage. The best results are highlighted with **bold**.

| | | Multi-Document QA | | | Single-Document QA | | | Summarization | | | Few-shot Learning | | | Synthetic | | Code | | |
|---|---|---|---|---|---|---|---|---|---|---|---|---|---|---|---|---|---|---|
| | Method | HotpotQA | 2WikiMQA | Musique | MF-en | NrtvQA | Qasper | GovReport | QMSum | MultiNews | TriviaQA | SAMSum | TREC | PRe | PCount | Lcc | RB-P | Avg. |
| **Llama-2-7B-Chat** | Full Cache | 30.20 | 27.37 | 11.54 | 34.33 | 19.68 | 19.40 | 24.60 | 20.81 | 26.19 | 84.61 | 41.00 | 63.50 | 8.00 | 4.50 | 61.46 | 55.14 | 33.27 |
| | StreamingLLM | 24.41 | 26.90 | 7.70 | 17.44 | 13.69 | 15.07 | 16.73 | 19.21 | 18.25 | 81.67 | 35.03 | 39.50 | 6.00 | 4.00 | 55.22 | 51.00 | 26.99 |
| | H2O | 29.92 | 25.07 | 10.46 | 23.05 | 17.04 | **18.45** | 19.96 | 20.07 | 24.12 | 82.66 | 38.26 | 59.50 | 3.50 | 4.00 | 56.01 | 51.64 | 30.23 |
| | CAM | 24.68 | 26.80 | 7.57 | 16.79 | 14.11 | 15.98 | 16.57 | 19.09 | 18.35 | 81.09 | 35.17 | 39.50 | 5.50 | 4.00 | 55.34 | 51.15 | 26.98 |
| | SnapKV | 28.96 | 26.20 | 11.14 | 28.41 | 15.72 | 17.10 | 17.01 | 20.30 | 21.53 | 84.32 | 38.62 | 57.00 | 8.50 | 4.00 | 57.26 | 53.03 | 30.57 |
| | EMS(w.o. pos) | **31.39** | 27.46 | 10.48 | **30.45** | **18.52** | 17.87 | **22.18** | 20.82 | 22.30 | 84.86 | 39.34 | **61.00** | 6.00 | **5.00** | 57.86 | **54.10** | 31.85 |
| | EMS(w. pos) | 30.93 | **27.58** | **11.37** | 29.85 | 17.49 | 17.17 | 18.59 | 20.57 | 22.05 | **85.60** | 38.79 | 60.50 | **9.50** | 4.50 | **58.90** | 53.55 | 31.68 |
| **Llama-3-8B-Instruct** | Full Cache | 44.93 | 37.92 | 24.10 | 41.89 | 22.84 | 39.26 | 28.69 | 23.57 | 26.58 | 90.31 | 42.67 | 74.50 | 67.00 | 6.48 | 57.13 | 51.34 | 42.45 |
| | StreamingLLM | 37.64 | 25.25 | 16.94 | 22.61 | 16.58 | 18.73 | 18.42 | 20.15 | 19.25 | 78.43 | 39.29 | 53.00 | 65.08 | 7.25 | 56.83 | 53.72 | 34.32 |
| | H2O | 43.74 | 34.06 | 20.65 | 27.71 | 20.34 | 26.67 | 21.27 | 20.58 | 23.98 | 88.52 | 38.50 | 58.50 | 66.50 | **7.50** | 57.14 | 52.11 | 37.99 |
| | CAM | 37.64 | 25.19 | 16.94 | 22.64 | 16.58 | 18.58 | 18.49 | 20.10 | 19.24 | 78.43 | 39.34 | 53.00 | 65.08 | 7.25 | 56.83 | 53.56 | 34.31 |
| | SnapKV | 41.96 | 31.82 | 20.06 | 34.91 | 20.52 | 26.59 | 19.94 | 21.72 | 21.94 | **90.47** | 39.55 | 50.50 | **67.00** | 6.84 | 58.23 | 53.81 | 37.87 |
| | EMS(w.o. pos) | **45.51** | **36.86** | **22.80** | **37.31** | **21.33** | 27.74 | **22.11** | **22.28** | **24.05** | 90.44 | 40.26 | **64.50** | 66.67 | 7.22 | 57.01 | 50.37 | **39.78** |
| | EMS(w. pos) | 44.09 | 32.70 | 21.77 | 35.67 | 20.84 | 26.87 | 21.71 | 21.89 | 23.89 | 89.75 | 39.96 | 61.00 | **67.00** | 7.24 | **60.51** | **56.46** | 39.40 |
| **LongChat-7B-v1.5-32k** | Full Cache | 31.26 | 22.52 | 12.16 | 43.92 | 18.33 | 28.81 | 31.29 | 22.57 | 26.32 | 82.41 | 40.06 | 66.00 | 31.50 | 0.00 | 53.02 | 55.28 | 35.34 |
| | StreamingLLM | 13.06 | 9.62 | 1.97 | 4.48 | 4.02 | 8.64 | 1.00 | 3.18 | 0.92 | 4.72 | 6.96 | 20.50 | 2.96 | 0.50 | 2.16 | 5.93 | 5.66 |
| | H2O | 11.33 | 10.37 | 1.20 | 7.13 | 4.04 | 8.59 | 2.42 | 9.52 | 2.92 | 23.26 | 7.11 | 27.50 | 2.45 | 0.15 | 9.30 | 8.24 | 8.47 |
| | CAM | 13.40 | 9.17 | 1.82 | 5.07 | 4.08 | 8.62 | 1.01 | 3.24 | 0.79 | 4.23 | 7.12 | 21.50 | 3.55 | 0.50 | 2.05 | 5.98 | 5.76 |
| | SnapKV | 19.60 | 13.45 | 8.39 | 15.06 | 9.07 | 9.19 | 3.76 | 13.15 | 3.42 | 57.43 | 18.26 | 30.50 | 1.00 | 0.44 | 15.60 | 20.08 | 14.90 |
| | EMS(w.o. pos) | **32.07** | 23.75 | **12.34** | **39.93** | 16.07 | 22.96 | **23.89** | **21.22** | **23.49** | **79.58** | **36.89** | 61.50 | 11.50 | 0.00 | **55.08** | 50.78 | 31.94 |
| | EMS(w. pos) | 31.51 | **23.75** | **12.34** | 37.16 | **16.39** | **23.47** | 20.33 | 21.18 | 22.83 | 78.70 | 36.76 | **61.50** | 27.75 | **0.60** | 54.02 | **52.39** | **32.54** |
| **Mistral-7B-Instruct-v0.2** | Full Cache | 36.42 | 21.77 | 19.13 | 47.12 | 21.02 | 29.62 | 32.57 | 24.02 | 27.09 | 86.23 | 42.99 | 71.00 | 89.33 | 3.07 | 54.00 | 51.87 | 41.08 |
| | StreamingLLM | 21.60 | 13.24 | 10.25 | 26.35 | 13.72 | 11.58 | 17.98 | 19.73 | 18.92 | 80.67 | 40.26 | 50.50 | 24.80 | **3.82** | 50.57 | 44.04 | 28.00 |
| | H2O | 23.31 | 14.07 | 10.32 | 33.42 | 14.29 | 16.75 | 23.12 | 21.09 | 23.73 | 83.22 | 38.74 | 63.00 | 31.50 | 3.44 | 49.17 | 44.66 | 30.86 |
| | CAM | 21.15 | 13.25 | 10.25 | 26.41 | 13.72 | 11.6 | 17.88 | 19.72 | 18.94 | 80.7 | 40.19 | 50.5 | 25.05 | 3.82 | 50.6 | 44.04 | 27.99 |
| | SnapKV | 26.28 | 14.58 | 12.02 | 40.82 | 16.88 | 18.87 | 21.46 | 21.60 | 22.05 | 84.82 | **40.79** | 51.00 | **70.08** | 2.72 | 51.01 | 46.75 | 33.86 |
| | EMS(w.o. pos) | 25.83 | 15.36 | 12.38 | 40.99 | **17.44** | 18.84 | **23.13** | 22.21 | **24.07** | 84.94 | 40.28 | 64.00 | 60.77 | **4.13** | 51.87 | 47.59 | 34.61 |
| | EMS(w. pos) | **26.49** | **15.91** | **12.57** | **41.51** | 17.28 | 18.90 | 22.63 | **22.50** | 23.82 | **85.17** | 40.44 | 63.50 | 69.40 | 3.74 | 51.49 | 46.88 | **35.14** |

## 5.3 Needle-in-a-Haystack

Needle-in-a-Haystack (Kamradt, 2023) is a challenging task to assess the model's ability to retrieve specific information from a large volume of data. We test our method using Mistral-7B-Instruct-v0.2 with 10 depths, 40 lengths and maximum token limit 32k. The merge threshold $\tau$ and merge magnification factor $\gamma$ are set to 0.55 and 4. As shown in Table 2, EMS delivers the best retrieval performance across all three compression budgets. It retains 95.9% retrieval ability of the fully cached model, even surpassing SnapKV, which is specifically designed for retrieval tasks. The visualization of retrieval accuracy is shown in Appendix D.

Table 2: The performance of Needle-in-a-Haystack across three budgets.

| $N_{budget}$ | 128 | 256 | 512 |
|---|---|---|---|
| H2O | 0.312 | 0.335 | 0.384 |
| SnapKV | 0.802 | 0.893 | 0.956 |
| EMS | **0.818** | **0.896** | **0.959** |

To fully test the performance of EMS, we also evaluate the language modeling ability on PG19 (Rae et al., 2019) and consistently achieving the lowest perplexity, which is shown in Appendix C.

## 5.4 Efficiency Analysis

**Time analysis**. To assess the efficiency of EMS, we measured the end-to-end latency on two RTX 4090 GPUs. As shown in Table 3, by compressing the KV cache, EMS supports larger batch sizes, whereas the fully cached model encounters out-of-memory (OOM) errors with batch sizes of 2 or 4. Moreover, the efficiency gains of our method become more pronounced with larger batch sizes, leading to increased throughput as batch size grows. For example, with 4096 prompt tokens and a generated token length of 8192, our method achieves a $6.74\times$ improvement in throughput compared to the fully cached model.

**Memory analysis**. EMS compresses the storage overhead for the full KV cache $dN_{full}$ to a constant $dN_{budget}$ per head. The extra static memory overhead is required for $s_{Glo}$, $s_{Loc}^{past}$, $s_{Loc}^{cur}$, $\|K\|$ and the mapping look-up-table, all of which are vectors with the maximum length $\gamma N_{budget}$. The extra memory is $5\gamma N_{budget}/(dN_{full})$ compared to full cache baseline. Considering $\gamma = 4, N_{budget} = 256, N_{full} = 4096, d = 128$, the metadata overhead is determined to be a mere 0.97%. During the computation time of certain layer, only $N_{budget}$ key-value states are loaded and expanded to $\gamma N_{budget}$ as runtime KV cache, providing more contexts.

Table 3: Comparisons of the end-to-end latency (s). The cache budget is set to 256. With constant budget, EMS supports larger batch size and generalize to longer context without OOM errors.

| $L_{prompt} + L_{gen}$ | Method | Batch Size | | | | | Max Throughput |
|---|---|---|---|---|---|---|---|
| | | 1 | 2 | 4 | 8 | 16 | (tokens/s) |
| 128 + 4096 | Full Cache | 464 | 799 | OOM | OOM | OOM | 10.3 |
| | EMS | 435 | 483 | 595 | 925 | 1622 | 40.4 (3.94×) |
| 128 + 6144 | Full Cache | 955 | 1685 | OOM | OOM | OOM | 7.3 |
| | EMS | 673 | 781 | 963 | 1552 | 2757 | 35.7 (4.89×) |
| 4096 + 4096 | Full Cache | 463 | 800 | OOM | OOM | OOM | 10.2 |
| | EMS | 441 | 486 | 604 | 943 | 1661 | 39.5 (3.85×) |
| 4096 + 8192 | Full Cache | 1589 | OOM | OOM | OOM | OOM | 5.2 |
| | EMS | 878 | 1100 | 1265 | 2082 | 3772 | 34.7 (6.74×) |

## 5.5 ABLATION STUDY

We ablate the effectiveness of the Global-Local score and the Evict-then-Merge strategy, and evaluate the performance scalability across different cache budgets $N_{budget}$ using the LongBench average score on Llama-2-7B-Chat.

**Ablation on Token Selection and Compression Strategy.** Table 4 illustrated the effectiveness of our method. By incorporating the Evict-then-Merge strategy, the average score improves 0.61 points compared to the evict-only approach. Furthermore, integrating both global and local scores increases the average score by 0.37 and 2.62 points, respectively, compared to using either the global or local score alone. These results highlight the effectiveness of our method in token selection and compression strategy.

Table 4: KV cache compression using different token importance scores and compression strategies.

| Score + Compression | Avg. |
|---|---|
| $s_{Glo}$ + Evict-then-Merge | 28.81 |
| $s_{Loc}$ + Evict-then-Merge | 31.06 |
| $s_{Glo-Loc}$ + Evict-only | 30.82 |
| $s_{Glo-Loc}$ + Evict-then-Merge | 31.34 |

**Cache Budgets.** In Table 5, we explore the impact of different cache budgets. The results indicate that even under extreme compression settings, performance remains relatively stable. Additionally, performance scales with cache size, approaching that of the fully cached model as the budget increases. When $N_{budget} = 1024$, EMS performs a difference of 0.27 from the full cache model. More results on different LLMs can be seen in Table 9.

Table 5: The impact of different cache budgets for each head. The performance scales with the budgets and reaches the performance of full cache baseline when the budget reaches a certain size.

| $N_{budget}$ | 128 | 256 | 512 | 768 | 1024 |
|---|---|---|---|---|---|
| Avg. | 29.41 | 31.34 | 32.26 | 32.71 | 33.00 |

Besides, we conduct the ablation and analysis on merge threshold $\tau$ and merge magnification factor $\gamma$, which affect merge-evict ratio and merge size, which are shown in Appendix B.

## 6 CONCLUSION

In this paper, we propose EMS, an input-aware, head-wise efficient KV cache management framework. By leveraging the proposed Global-Local attention score, EMS addresses the biased distribution of important KV tokens caused by accumulated attention scores and positional encoding, leading to a more balanced selection of tokens. We further design a unified Evict-then-Merge strategy based on the redundancy and sparsity intrinsic of KV tokens across different heads. In particular, we implement a zero-class mechanism to enable parallel computation for head-wise operation. Extensive experiments on language modeling perplexity, LongBench, and Needle-in-a-Haystack tasks demonstrate the SOTA performance, validating the effectiveness of EMS.

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

# A    METHOD DETAILS AND DISCUSSION

**Two-Levels of Parallelism.** We regarded EMS as a parallel solution for two reasons. On the one hand, as illustrated in Section 4.1, the sparsity and redundancy provide the potential for more extreme compression. However, the head-wise characteristic can lead to different cache budgets for different heads. For code implementation, the shape of KV states is (`batch_size`, `num_heads`, `kv_len`, `head_dim`), which means that all heads are concatenated as a tensor and computed in parallel. Luckily, there are two properties and they exhibit complementarity to some extent. For example, we can evict more for the heads with higher sparsity and less redundancy (dark blue and light red bars), and merge more for the heads with higher redundancy and less redundancy (light blue and dark red bars). So we take this chance to allocate the same budget for each head. On the other hand, merge and evict are two different operations. Different heads may have different operations and different merge-evict ratios. To unify the merge and evict process, we introduce a zero-class, thus the eviction can be treated as merging, and we only do merging at the second stage of Evict-then-Merge strategy.

**Two Levels of Eviction.** Under our Evict-then-Merge strategy, there are two levels of eviction consideration, which is shown in Figure 4a. Under the long context scenarios, not all tokens are necessary for language modeling and can lead to significant memory overhead. Hence, the less relevant tokens will first be evicted and leaves $N_{imp} + N_{tbm} + N_{loc}$ tokens for subsequent merging, which is same for each heads. At the merge stage, considering that some TBM tokens cannot find a suitable merge destination for low similarity, evicting them can avoid disturbing the class centers. So these tokens are merged to zero-class, which is equivalent to eviction.

**Evict-then-Merge Details.** There are some implementation differences in the compression of prefilling and decoding stage. Specifically, at prefilling stage, numerous tokens are filled at once, which might far exceed the size of the cache budget. Therefore, it is necessary to partition them and determine the TBM tokens and class-center tokens. And this stage often requires to merge multiple tokens for each head. The merged $N_{imp}$ tokens serve as the class centers, where $N_{tbm} + N_{imp}$ tokens will share the $N_{imp}$ entries. At decoding stage, we focus more on the update of class centers and the mapping relation, which is shown in Figure 4b. To achieve dynamic class center, the merge operation is processed at the class center level, which means that merging will change the mapping of the TBM tokens and class centers. After each decoding in the auto-regressive generation, a new token will exceed local range and is regarded as the class center. One least important class center token is selected as TBM token. For each head, there are two kinds of operations for this TBM token. Different heads may have different decisions for merge or evict operation, and they are unified as merge. Since we keep the cache budget constant, the number of expanded tokens is also fixed. So we need to evict one element from the look-up-table and fill the new mapping.

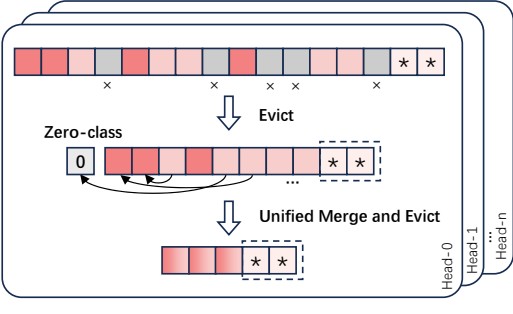
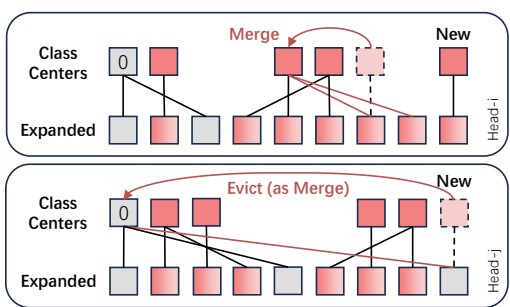

(a) Two levels of eviction.          (b) Unified merge and evict at decoding stage.

Figure 4: Evict-then-Merge details. (a) Two levels of eviction. The first level of eviction is evicting the same number of irrelevant tokens. The second level of merge is merging the tokens with low similarity to zero-class token. Different heads have different eviction at the second level. (b) Unified merge and evict at decoding stage. Different heads in the same layer have different merge or evict decisions, which are unified as merge operation.

**Shared Entry Expansion.** The motivation of expansion is that we hope to conduct clustering on the KV tokens based on cosine similarity. In this way, we can use a small number of class center tokens to represent more tokens. Therefore, those similar tokens share the same KV entry and need to be expanded during computation. For the implementation of expansion, we keep a position look-up-table and $N_{imp}$ tokens are expanded to $\gamma N_{imp}$ tokens at computation time of the certain layer. In this way, more context tokens participate in the computation, and only additional overhead is only brought to the current computed layer.

**Global Score Efficiency Discussion.** FlashAttention has become a standard technique for long sequence inference. It achieves more efficient attention computation through tiling, yet doesn't return the attention weights. The calculation of the global score requires obtaining all the attention weights. To be compatible with Accumulated FlashAttention, recent works have also made some optimizations. ZipCache (He et al., 2024) approximates global scores by sampling 10% probe tokens, which is orthogonal and compatible with our work. NACL (Chen et al., 2024a) has implemented the accumulated attention weights of FlashAttention by recomputation based on Paddle. Thus, the dilemma regarding the global score can be solved.

## B    ABLATION STUDY

**Merge Threshold.** The merge threshold plays a crucial role in determining whether a token will be merged into the zero class (i.e., evicted) or common class centers. A smaller threshold results in more tokens being evicted, while a larger threshold leads to more aggressive merging. Table 6 presents the results of varying the merge threshold, indicating that a moderate threshold $\tau = 0.6$, yields best performance on balance.

Two extreme cases are setting threshold to 0 or 1, meaning all merging or all evicting. If $\tau = 0$, all TBM tokens are merged without considering that some tokens are not suitable to merge, leading to 1.66 performance drop. On contrary, if $\tau = 1$, all TBM tokens are evicted, resulting in a performance degradation of 0.52. Therefore, both over merging and over evicting yields sub-optimal performance, manifesting the effectiveness of joint merge and evict.

Table 6: Effects of merge threshold $\tau$ on LongBench. The TBM tokens with a redundancy score above $\tau$ are merged, while those below are evicted. A lower threshold results in more tokens being merged.

| $\tau$ | 0 | 0.1 | 0.2 | 0.3 | 0.4 | 0.5 | 0.55 | 0.6 | 0.65 | 0.7 | 0.75 | 0.8 | 0.9 | 1 |
|---|---|---|---|---|---|---|---|---|---|---|---|---|---|---|
| Avg. | 29.68 | 29.59 | 29.84 | 30.25 | 30.69 | 30.93 | 30.88 | **31.34** | 31.28 | 31.25 | 31.27 | 30.49 | 31.13 | 30.82 |

**Merge Size.** In our evict-then-merge strategy, the number of tokens retained for class center and merging (i.e $\gamma N_{budget}$) after the initial eviction stage is also a key factor, which is indicated by merge magnification factor $\gamma$. As shown in Table 7, we ablate the effect of $\gamma$ and find that merging approximately $3 \sim 4$ times of the class center size achieve the optimal performance.

Two extreme cases are setting $\gamma$ to 1 or Inf, which means only evict irrelevant tokens and no eviction of irrelevant tokens. When $\gamma = 1$, we only apply the first level of eviction and skip the unified merge and evict operation. On the contrast, when $\gamma = $ Inf, we skip the eviction of irrelevant tokens and only implement the unified merge and evict at a longer token length. The results demonstrate that both over-merging and under-merging can negatively impact the performance. Removing some irrelevant tokens at an appropriate degree not only improves performance, but also reduces the complexity of managing the mapping relation.

Table 7: Ablation on $\gamma$, which affects the merge size. 'Inf' means there is no eviction of irrelevant contexts and '1' means only apply eviction using $\boldsymbol{s}_{Glo-Loc}$.

| $\gamma$ | 1 | 2 | 3 | 4 | 5 | 6 | 7 | 8 | Inf |
|---|---|---|---|---|---|---|---|---|---|
| Avg. | 30.05 | 30.88 | 31.00 | **31.34** | **31.34** | 31.25 | 31.04 | 30.92 | 31.18 |

**Key Similarity, Value Similarity and Key-Value Similarity.** We conduct experiments to evaluate the performance of merging based on key, value and key-value similarity. The results arc displayed in Table 8, which manifest the superiority of considering both key similarity and value similarity.

Table 8: The comparison of using different similarities. The performance metric is the average score on LongBench using Llama2.

| $Similarity$ | Key | Value | Key-Value |
|---|---|---|---|
| Avg. | 31.18 | 31.26 | **31.34** |

**Cache Budgets.** More results on the LongBench performance of different models using more budgets and their comparisons to baselines are shown in Table 9. $\gamma$ and $\tau$ are set to 4 and 0.6 as ablated. EMS outperforms other methods on the 4 LLMs across different cache budgets.

Table 9: Impact of varying cache budget across four LLMs. EMS outperforms other methods on the 128, 256, 512, 768 and 1024 cache budgets.

| | Method | 128 | 256 | 512 | 768 | 1024 |
|---|---|---|---|---|---|---|
| **Llama-2** | StreamingLLM | 24.39 | 26.99 | 28.92 | 29.46 | 29.77 |
| | H2O | 19.00 | 30.23 | 31.14 | 31.54 | 31.91 |
| | SnapKV | 27.78 | 30.57 | 31.64 | 32.05 | 32.27 |
| | EMS(w.o. pos) | **29.41** | **31.34** | **32.26** | **32.71** | **33.00** |
| | EMS(w. pos) | 29.12 | 30.93 | 31.99 | 32.53 | 32.68 |
| **Llama-3** | StreamingLLM | 32.42 | 34.32 | 36.18 | 37.27 | 38.80 |
| | H2O | 36.02 | 37.99 | 39.55 | 40.20 | 40.62 |
| | SnapKV | 35.54 | 37.87 | 39.78 | 40.57 | 40.95 |
| | EMS(w.o. pos) | **36.45** | **38.94** | **40.45** | **40.74** | **41.12** |
| | EMS(w. pos) | 36.14 | 38.62 | 40.12 | 40.73 | 41.10 |
| **LongChat** | StreamingLLM | 5.77 | 5.66 | 5.45 | 8.51 | 13.98 |
| | H2O | 5.71 | 8.47 | 17.71 | 26.25 | 31.19 |
| | SnapKV | 6.04 | 14.90 | 32.08 | 33.27 | 33.63 |
| | EMS(w.o. pos) | **25.65** | **31.25** | 33.00 | 33.05 | 33.36 |
| | EMS(w. pos) | 18.85 | 31.10 | **33.32** | **34.18** | **34.01** |
| **Mistral** | StreamingLLM | 26.97 | 28.00 | 29.90 | 30.74 | 31.37 |
| | H2O | 29.89 | 30.86 | 32.02 | 33.38 | 34.24 |
| | SnapKV | 31.15 | 33.86 | 36.16 | 37.24 | 37.91 |
| | EMS(w.o. pos) | 31.16 | 33.93 | 35.37 | 36.30 | 37.14 |
| | EMS(w. pos) | **31.44** | **34.66** | **36.59** | **37.64** | **38.35** |

## C  LANGUAGE MODELING PERPLEXITY

We evaluate the perplexity on the PG19 (Rae et al., 2019) dataset using the LLaMA-2-7B model. The experiment is conducted under three cache budgets: 2%, 5%, and 10% of the pretraining length (i.e. 4096 for LLaMA-2-7B). The testing sequence length is extended 10 times of the original size, allowing us to thoroughly assess the model's continuous modeling capability for long contexts. The merge threshold $\tau$ and merge magnification factor $\gamma$ are set to 0.6 and 4. No protection is applied, allowing more aggressive merging.

As shown in Figure 5, EMS consistently maintains the lowest perplexity across all budget settings. This demonstrates its robustness in managing limited cache resources while preserving language modeling accuracy. The fully cached baseline encounters significant degradation in performance once the sequence length exceeds the pretraining limit.

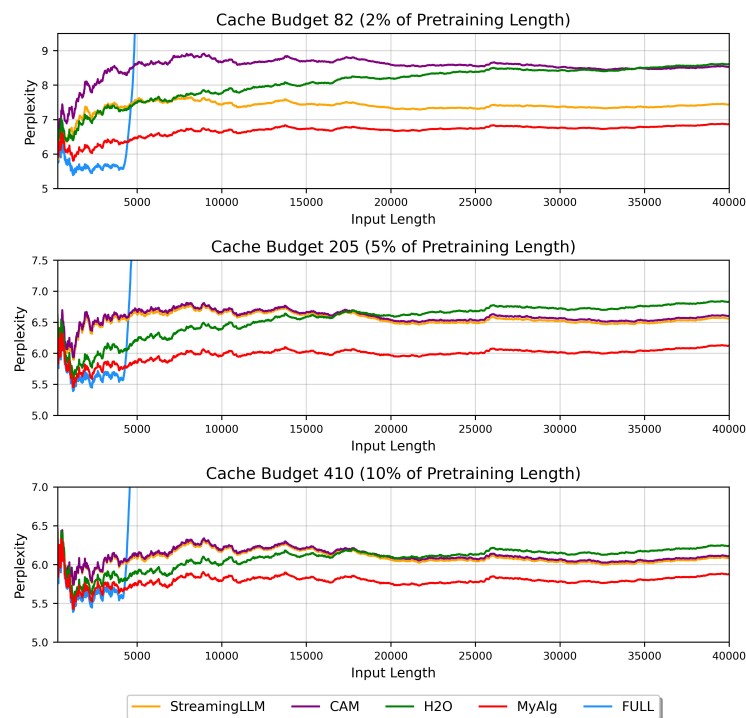

Figure 5: Perplexity across different cache budgets. Lower perplexity indicates better model performance. SnapKV is not listed for its lack of compression in decoding stage.

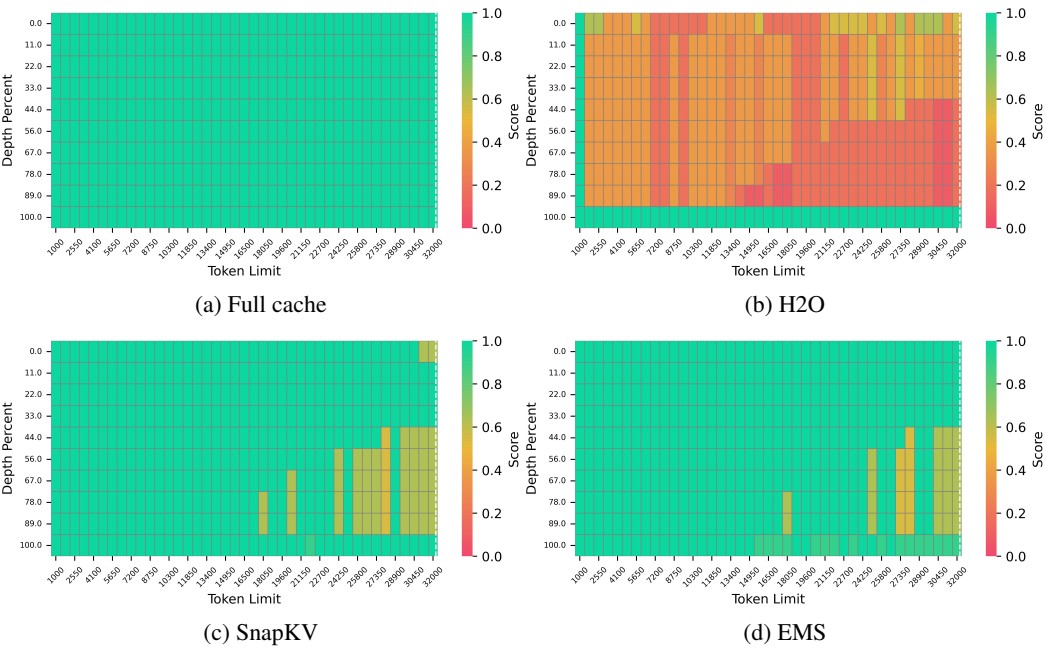

Figure 6: Pressure testing results on Mistral-7B-Instruct-v1.5 with full cache and three compression methods. The maximum test length is 32k, which is almost max context length for popular LLMs. For compression methods, the cache budget for each head is 512, which is only 1.6% of the maximum testing length.

## D   NEEDLE-IN-A-HAYSTACK VISUALIZATION

Needle-in-a-Haystack is particularly challenging as it requires precise retrieval from extensive context, simulating real-world scenarios where relevant information is buried among irrelevant data. Table 2 has shown the numerical results, here we visualize the comparisons of baselines on retrieval accuracy across the depths and token limits under budget 512, which is shown in Figure 6. For EMS, the merge threshold $\tau$ and merge magnification factor $\gamma$ are set to 0.6 and 4.

We can see that H2O almost collapses on retrieval task, with the accuracy only 38.4%. SnapKV, a method designed for retrieval tasks, achieves 95.6% accuracy, while EMS can achieve 96.8% retrieval ability of the fully cache model.

## E   TOKEN SELECTION ACROSS LAYERS

In Figure 8, we visualize more token selection patterns of different methods across layers using Llama2 and Llama3. The experiment is conducted on multi-document QA dataset *hotpotqa* (Yang et al., 2018) and summarization dataset *gov_report* (Huang et al., 2021). The heads of Llama3 are expanded 4 times due to the use of GQA (Ainslie et al., 2023).

To further analysis the how tokens are selected according to Global-Local score, we draw the proportion of selected tokens derived from global-aware, local-aware and local-only selection in Figure 7. The token numbers are averaged on head dimension. If selecting Top-256 from 4096 tokens, the number of tokens sourced from global-aware and local-aware sources is roughly 1:1. If selecting Top-1024 from 4096 tokens, this ratio is approximately 7:3. This implies that when merging 1024 tokens into 256 tokens, more global-aware tokens are merged. Since local tokens are changing during the decoding stage, our dynamic class centers can be indispensable.

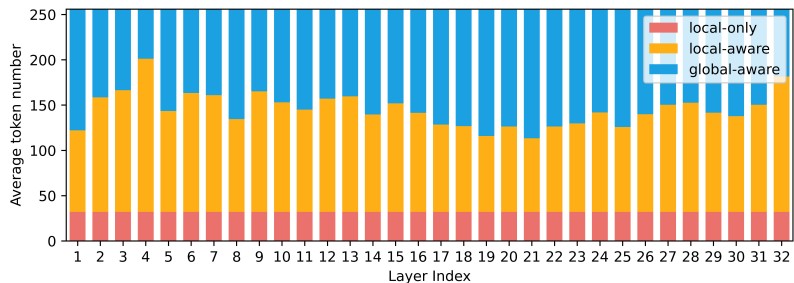

(a) Select Top-256 out of 4096 tokens.

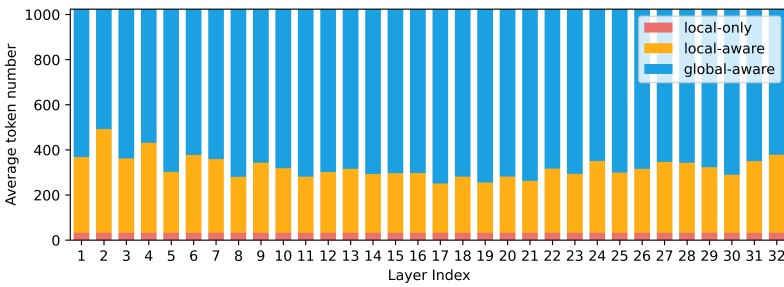

(b) Select Top-1024 out of 4096 tokens.

Figure 7: The distribution of selected tokens. The sample is taken from the *gov_report* dataset. When selecting fewer tokens, the Global-Local based selection method tends to evenly choose between global-aware and local-aware tokens. And when selecting more tokens, it leans toward selecting more global-aware tokens.

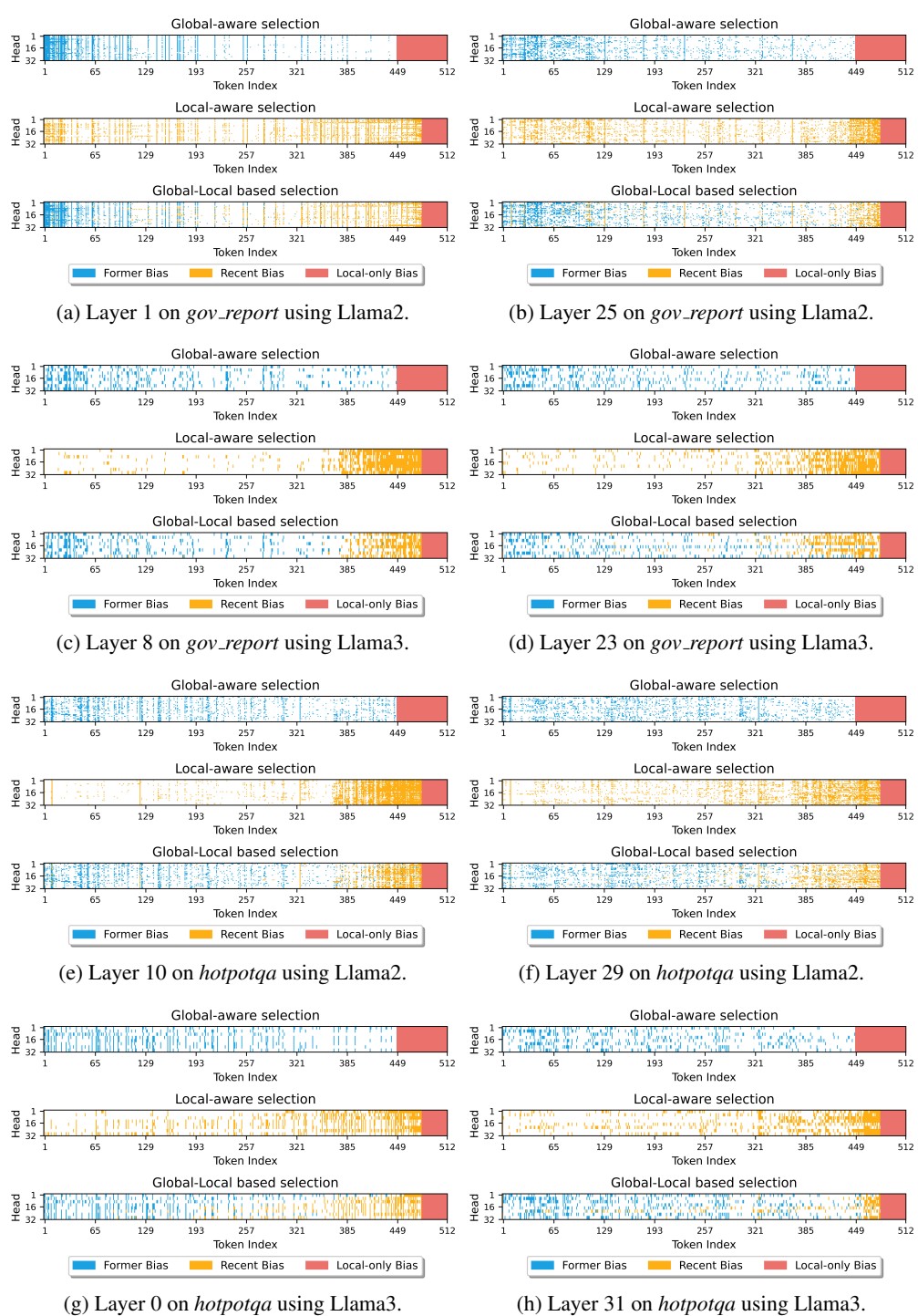

Figure 8: Token selection patterns visualization on different layers, datasets and models. Global-aware selection tends to select former tokens while local-aware selection prefers recent tokens. Global-Local based selection can balance these two approaches.