# OpenReview forum: "EMS: Adaptive Evict-then-Merge Strategy for Head-wise KV Cache Compression Based on Global-Local Importance"
_ICLR.cc/2025/Conference — Submitted to ICLR 2025_

### Official Review · Reviewer_kiKQ · 2024-10-25

**Soundness:** 2
**Presentation:** 2
**Contribution:** 1
**Rating:** 5
**Confidence:** 4

**Summary:**

The authors propose a head-wise adaptive strategy with a similarity-based merge method and a global-local scoring mechanism to improve KV cache compression performance under extreme compression ratios.

**Strengths:**

The authors propose joint optimization strategies in three directions to improve cache compression.

**Weaknesses:**

1. Evaluation is conducted only under a budget of 256, which is insufficient to account for data fluctuations. The method should be evaluated across a range of budgets (e.g., 128, 256, 512) to provide robust conclusions. Results for varying budgets in Table 5 lack sufficient validity without baseline comparisons, as all cache compression methods approach the full cache baseline with increased budgets. Including comparisons with baselines under different budget conditions would be more convincing.
2. More details regarding how the baseline value of 31.43 was derived in ablation study should be provided, especially since this value is not referenced in Table 1. Additionally, more ablation results, such as comparisons under different budget settings and using varied models, should be provided to ensure a comprehensive evaluation.
3. Results deviate significantly from prior work under full cache conditions, which may indicate potential issues in the experimental setup or implementation. On the synthetic PRe dataset, all models score close to 0 in this paper, while prior works [1-2] reported scores of 86.98 and 89.3 for Mistral-7B, and both 30.5 for LongChat-7B. A thorough investigation into these discrepancies is recommended
4. The authors propose head-wise adaptivity, similarity based cache merging, and a refined attention score for enhancing cache compression. However, some works exist for each of these aspects[3-5]. The authors should clarify differences from existing research.
5. Given FlashAttention has become a standard technique for long sequence inference, using a global score may not be practical due to the additional quadratic computation and storage overhead needed to recompute global attention weights. In contrast, the local score only reconstructs attention weights for recent tokens, avoiding this burden. Thus, a global score should only be used if it delivers significant benefits; otherwise, the local score is more practical.

[1]Li, Yuhong, et al. "Snapkv: Llm knows what you are looking for before generation."

[2]Yuan, Jiayi, et al. "Kv cache compression, but what must we give in return? a comprehensive benchmark of long context capable approaches."

[3]Feng, Yuan, et al. "Ada-KV: Optimizing KV Cache Eviction by Adaptive Budget Allocation for Efficient LLM Inference."

[4]Wang, Zheng, et al. "Model tells you where to merge: Adaptive kv cache merging for llms on long-context tasks."

[5]Chen, Yilong, et al. "Nacl: A general and effective kv cache eviction framework for llms at inference time."

**Questions:**

Please refer to the weaknesses.

---

> ### Author Response · Authors · 2024-11-22
> **Response to Reviewer kiKQ. W1, W2, W3**
>
> Thank you for taking the time to thoroughly read our article and for pointing out its shortcomings. Here is our response:
>
> &nbsp;
>
> # W1
>
> We have supplemented the performance of different models using more budgets and their comparisons to baselines. We set $\gamma=4$, $\tau=0.6$ as ablated in Appendix B.
>
> | Method                 | 128     | 256      | 512     | 768      | 1024  |
> | ---- | ---- | ---- | ---- | ---- | ---- |
> | StreamingLLM | 24.39 | 26.99 | 28.92 | 29.46 | 29.77 |
> | H2O                       | 19.00 | 30.23 | 31.14 | 31.54 | 31.91 |
> | SnapKV                 | 27.78 | 30.57 | 31.64 | 32.05 | 32.27 |
> | EMS                        | **29.41** | **31.34** | **32.26** | **32.71** | **33.00** |
>
> More results can be seen in **Table 9**. Through the results, we can see that EMS outperforms other methods on the 4 LLMs across different cache budgets.
>
> &nbsp;
>
> # W2
>
> In the ablation part, the parameters are fixed to the best parameters obtained by the ablation ($\gamma=4$, $\tau=0.6$). In Table 1, we conducted multiple sets of tests on a subset of parameters ($\gamma=3:5$, $\tau=0.5:0.7:0.05$) around the best parameters and reported the best results. Here, we also provide the mean performance on this subset.
>
> | Method                 | Llama2 | Llama3 | LongChat | Mistral |
> | ---- | ---- | ---- | ---- | ---- |
> | StreamingLLM | 24.39    | 34.32    | 5.66             | 28.00  |
> | H2O                       | 19.00    | 37.99    | 8.47             | 30.86  |
> | SnapKV                 | 27.78    | 37.87    | 14.90         | 33.86  |
> | EMS(w.o. pos)   | **31.19**    | **38.97**     | 31.40         | 33.84  |
> | EMS(w. pos)       | 30.96     | 38.62    | **31.47**          | **34.60**  |
>
> The experimental results show that even when using the mean results on the test subset, that is, without much parameter tuning, the performance of EMS still outperforms baselines.
>
> &nbsp;
>
> # W3
>
> Thanks for pointing out this problem! This is not the problem of experiment pipeline. We checked the table and find that we incorrectly set the table header when organizing the table, which led to the biases in the reported scores. We have updated the table header and did not change any data. The results now can match.

---

> ### Author Response · Authors · 2024-11-22
> **Response to Reviewer kiKQ. W4, W5**
>
> # W4
>
> **Ada-KV[3]** is a **cache budget allocation** method that tries to allocate different budgets across heads. EMS uses uniform cache budget for each head, which is orthogonal and compatible to the head-wise cache budget work.
>
> The similar part between **KVMerger [4]** and EMS is that we both do KV Cache merging. However, they differs in merging mechanism and compression strategy. For the **merging mechanism**, EMS utilizes a cross-position strategy, where KVMerger is to merge adjacent tokens. For the **compression strategy**, EMS designs both eviction and unified merging operation, while KVMerger only do merging.
>
> **Nacl[5]** is an eviction-based method that uses proxy tokens to accumulate the attention weights. EMS **decouples the compression into 2 steps: identifying token importance and applying evict-then-merge strategy**, and focuses more on the evict-then-merge strategy. Therefore, Nacl is generally orthogonal to EMS. Although they have also designed an accumulate pattern, which is a more intricate token importance indicator, their score can also be applied to EMS.
>
> [1] Feng, Yuan, et al. "Ada-KV: Optimizing KV Cache Eviction by Adaptive Budget Allocation for Efficient LLM Inference."
>
> [2] Wang, Zheng, et al. "Model tells you where to merge: Adaptive kv cache merging for llms on long-context tasks."
>
> [3] Chen, Yilong, et al. "Nacl: A general and effective kv cache eviction framework for llms at inference time."
>
> &nbsp;
>
> # W5
>
> From the perspective of identifying the token importance, **both global and local score have certain advantages**. As can be seen from Table 1, H2O and SnapKV perform well on different tasks. And our global-local score can adaptively combine them.
>
> Although the current calculation of global scores is not compatible with FlashAttention, there is potential for optimization in this area:
>
> - For instance, [1] can **approximate global scores through sampling**, which is orthogonal to and compatible with our approach.
> - FlashAttention itself is evolving. We are currently working on **improving its kernel to enable it to return accumulated attention scores**, which will further accelerate inference. Nevertheless, this will not change the nature of our method. Furthermore, we find that the recent work [2] has implemented accumulated attention weights of FlashAttention by recomputation based on Paddle. So the dilemma of the global score can be solved.
>
> [1] He, Yefei, et al. "ZipCache: Accurate and Efficient KV Cache Quantization with Salient Token Identification."
>
> [2] Chen, Yilong, et al. "Nacl: A general and effective kv cache eviction framework for llms at inference time."

---

> > ### Comment · Reviewer_kiKQ · 2024-11-24
> >
> > Thanks for your response. The clarification regarding W3 addressed my concerns about the correctness of the results, which was the primary reason for my initial score of 3. However, the issue of the computational efficiency of the global score remains unresolved, as the authors have not provided any evaluation to address this concern. I will raise my score to 5, but I still remain concerned about the computational efficiency of the global score.

---

> > > ### Author Response · Authors · 2024-11-24
> > > **Response to Reviewer kiKQ.**
> > >
> > > Thanks for your valuable suggestions and timely reponses, which contributes to the improvement of EMS. Integrating the FlashAttention kernel is exactly what we will do next, which will further accelerate the calculation of the global score.

---

### Official Review · Reviewer_emot · 2024-11-01

**Soundness:** 2
**Presentation:** 2
**Contribution:** 1
**Rating:** 5
**Confidence:** 5

**Summary:**

The proposed EMS includes:
1. the Global-Local score is designed for important token selection;
2. Evict-then-Merge compression strategy is designed;
3. an efficient head-wise parallel compression is designed for the KV cache.

EMS achieves extreme compression, retaining 95% retrieval accuracy using under 2% of the context length in the Needle-in-a-Haystack task.

**Strengths:**

The combination of local and global method is used in EMS, which compress the KV cache with a high compression rate and preserve the major context information.

**Weaknesses:**

The EMS depends on the online calculation of attention score and dynamic sparsity, leading to a high cost for practical deployment. And some sub-methods are similar to streamingLLM and H2O.

**Questions:**

We should explore the essential mechanism of the combination of local and global attention.

---

> ### Author Response · Authors · 2024-11-22
> **Response to Reviewer emot.**
>
> Thanks for your comments on the design of Global-Local score. Here is our response for your concernings:
>
> &nbsp;
>
> # W1
> The design of the compression strategy is essentially a **trade-off among performance, memory, and time**. The goal of EMS is to achieve the extreme compression of tokens, reduce memory overhead, and minimize the performance degradation as much as possible, which is of great significance for some long-context scenarios. Although EMS has some additional overhead, it can bring benefits in terms of performance and memory.
>
> About EMS, perhaps you have some misunderstanding of our approach. StreamingLLM and H2O are two pioneering works in KV cache eviction. Unlike them, EMS decouples the compression process into two phases: identifying token importance and applying evict-then-merge strategy. In the first phase, we found the flaws of StreamingLLM and H2O and designed a better token indicator to alleviate the biases they exhibit. In the second phase, we achieved more extreme compression based on the token importance. Therefore, **the first phase of EMS actually discovers the deficiencies of StreamingLLM and H2O and improves them**.
>
> &nbsp;
>
> # W2
>
> **Mechanism of the combination of local and global attention**:  By mean-alignment, EMS dynamically integrates the former- and local-biased scores and designs the more balanced Global-Local score to indentify the token importance more effectively. We investigate more on the relative weights of global and local scores and found that changing the relative weight has the potential to achieve better results. And **EMS can also achieve good performance when the relative weight is 1**. This easy-to-implement mean-alignment can improve the performance. In the future, we will continue to explore better token indicator, such as task-aware ratios.
>
> | relative weight | 0(loc)  | 0.1       | 0.25   | 0.33     | 0.5       | 1(glo-loc) | 2           | 3           | 4            | 5            | 10         | Inf(glo) |
> | ---- | ---- | ---- | ---- | ---- | ---- | ---- | ---- | ---- | ---- | ---- | ---- | ---- |
> | Llama2                | 31.06 | 31.02 | 31.00 | 31.08 | 31.27 | 31.34        | **31.49** | 31.37 | 31.19 | 31.15 | 30.71 | 28.81    |
> | Llama3                | 38.52 | 38.54 | 38.71 | 38.84 | 39.11 | 38.94        | 38.89 | 38.93 | 38.68 | 38.58 | 37.99 | 36.33 |
> | LongChat            | 29.32 | 29.56 | 29.94 | 30.28 | 30.45 | **31.25**        | 31.22 | 30.87 | 30.89  | 30.72 | 29.43 | 26.15 |
> | Mistral                  | 33.48 | 33.92 | 33.93 | 33.93 | 33.98 | 33.93        | 33.94 | **34.00** |  33.97 | 33.93 | 33.96 | 29.88 |

---

> ### Author Response · Authors · 2024-11-24
>
> Thank you for your valuable feedback on our submission. We greatly appreciate your efforts and constructive comments. As a kind reminder, the discussion period is drawing close. Please let us know if there remains anything that we can further clarify to improve our work. We understand that this is a busy period and appreciate your time and consideration.

---

> ### Author Response · Authors · 2024-12-01
>
> Thank you again for your contribution to the review process. With the clock ticking down on the discussion period, we kindly ask if you could revisit our submission, review our rebuttal, and share any additional remarks or conclusions you may have. Your continued engagement will ensure a more thorough and fair assessment of our work, which we believe holds significant potential contributions to the field.

---

### Official Review · Reviewer_h765 · 2024-11-03

**Soundness:** 2
**Presentation:** 2
**Contribution:** 2
**Rating:** 3
**Confidence:** 4

**Summary:**

The study proposes a new KV cache compression method, called EMS, in which there are 2 levels of compression. In the first stage, the proposed method evict unimportant tokens based on the proposed KV importance estimation method called Global-Local Score. The Global-Local Score is claimed to consider the importance of the token comprehensively, without biasing toward either previous or recent token, resulting in a more uniform selection of KV cache to retain. In the second stage, the proposed method perform Evict-the-Merge operation based on the measured sparsity and redundancy of KV cache among different heads.The conducted experiment shows the performance of the proposed method is better than other baselines, even under resource-intensive scenarios.

**Strengths:**

- The observation on KV similarity is interesting.

**Weaknesses:**

1. I'm not quite convinced with the redundancy estimation part, but maybe I misunderstood some parts here. What if both K & V of two tokens are highly dissimilar, i.e. cos_sim(K_x,K_y) = -1 and cos_sim(V_x,V_y) = -1, their product is now 1, which is deemed as redundancy by the proposed method. However, I believe this situation is not redundant at all since both K & V are highly dissimilar so we should keep both of them. Can the author clarify this to ease my concern?
2. The study only experimented on KV cache size = 256. However, I believe it would be more beneficial to see the proposed method's performance under different cache sizes. Can the authors provide more result regarding to this?
3. The "expansion" part in Evict-then-Merge plays an important role to implement the framework, but is not elaborated in the main text (or briefly in the appendix). Can the author elaborate on why and how we should do this expansion?
4. Can the authors make sure that the experiment pipeline is correct? For example, in [1] [2], the performance of Mistral full KV on MF-en, HotpotQA, 2Wiki, SAMSum are all around 49.34, 42.77, 27.35, 42.96 respective. However, in the submission's reported numbers, they are 29.62, 36.42, 21.77, 71.00 respectively, which are off by a large margin. I am doubting the experiment pipeline the authors used.
5. There are many variations of NIAH tasks, e.g. haystack formed from repetitive sentences or haystack formed from a long corpus. Can the authors elaborate which setting used in the study?

[1] PYRAMIDKV: DYNAMIC KV CACHE COMPRESSION BASED ON PYRAMIDAL INFORMATION FUNNELING, Arxiv, https://arxiv.org/abs/2406.02069
[2] DYNAMICKV: TASK-AWARE ADAPTIVE KV CACHE COMPRESSION FOR LONG CONTEXT LLMS, ICLR'25 submission, https://openreview.net/forum?id=uHkfU4TaPh

**Questions:**

Please see weaknesses.

---

> ### Author Response · Authors · 2024-11-22
> **Response to Reviewer h765. W1, W2**
>
> Thank you  for your careful reading of this paper and for pointing out issues in the details. Here is our response:
>
> &nbsp;
>
> # W1
>
> Thank you for pointing out our oversight. The case proposed by the reviewer is a minor one. We have retested the average performance on LongBench considering the absolute value of cosine similarity with a 256-cache budget. **Although the improvement has brought a certain performance gain, this gain is relatively small, and there is even a slight performance drop in some case**. Compared with the best baseline results, **the main increase is still brought by our method**. In the future, we will consider designing an adaptive method to make better use of similarity.
>
> | LLMs          | best baseline | before | after    |
> | ---- | ---- | ---- | ---- |
> | Llama2      | 30.57               | 31.34 | 31.43 |
> | Llama3      | 37.99               | 38.94 | 38.96 |
> | LongChat | 14.90               | 31.25 | 31.82 |
> | Mistral       | 33.86               | 33.93 | 33.92 |
>
> &nbsp;
>
> # W2
>
> Thank you for pointing out the shortcomings of the experiment. We have supplemented the performance results on LongBench across various cache budgets using different methods to make the evaluation of our approach more comprehensive.
>
> Here we show the performance on Llama2.
>
> | Method\Budget | 128     | 256      | 512     | 768      | 1024  |
> | ---- | ---- | ---- | ---- | ---- | ---- |
> | StreamingLLM | 24.39 | 26.99 | 28.92 | 29.46 | 29.77 |
> | H2O                       | 19.00 | 30.23 | 31.14 | 31.54 | 31.91 |
> | SnapKV                 | 27.78 | 30.57 | 31.64 | 32.05 | 32.27 |
> | EMS                        | **29.41** | **31.34** | **32.26** | **32.71** | **33.00** |
>
> More results can be seen in **Table 9**. Through the results, we can see that **EMS outperforms other methods on the 4 LLMs across different cache budgets**.

---

> ### Author Response · Authors · 2024-11-22
> **Response to Reviewer h765. W3, W4, W5**
>
> # W3
>
> **Motivation of expansion**: We hope to conduct clustering on the KV tokens based on cosine similarity. In this way, we can use a small number of class center tokens to represent more tokens. Therefore, those similar tokens share the same KV entry and need to be expanded during computation.
>
> **Implementation of expansion**: a position look-up-table is maintained and $N_{imp}$ tokens are expanded to $\gamma N_{imp}$ tokens at computation time of the certain layer.
>
> To validate the effectiveness of expansion, here we provide the comparison between the cases with and without expansion across different budgets on Llama2. We can infer that **the expansion can steadily bring performance gains**, which demonstrates its necessity.
>
> | Method                       | 128      | 256      | 512     | 768      | 1024  |
> | ---- | ---- | ---- | ---- | ---- | ---- |
> | with expansion        | **29.41** | **31.34** | **32.26** | **32.71** | **33.00** |
> | without expansion | 28.05 | 30.43 | 31.92 | 32.51 | 32.70 |
>
> &nbsp;
>
> # W4
>
> Thanks for pointing out this problem! This is not the problem of experiment pipeline. We checked the table and find that we incorrectly set the table header when organizing the table, which led to the bias in the reported scores. We have updated the table header without changing any data. The scores now can match.
>
> Besides, the implementation of EMS is partly based on the codebase of KIVI. Under our experiment pipeline, the performance of the model with full cache can match that of KIVI.
>
> [1] KIVI: A Tuning-Free Asymmetric 2bit Quantization for KV Cache, ICML’24, https://arxiv.org/abs/2402.02750.
>
> &nbsp;
>
> # W5
>
> We are using the haystack formed from a long corpus (Paul Graham Essays).

---

> ### Author Response · Authors · 2024-11-24
>
> Thank you for your valuable feedback on our submission. We greatly appreciate your efforts and constructive comments. As a kind reminder, the discussion period is drawing close. Please let us know if there remains anything that we can further clarify to improve our work. We understand that this is a busy period and appreciate your time and consideration.

---

> > ### Comment · Reviewer_h765 · 2024-11-24
> >
> > Thanks the authors for the rebuttal, but some of my concerns still remain:
> > - How can changing "absolute value of cosine similarity" address my question on 2 highly dissimilar KV pairs? If we have cos_sim(K_x,K_y) = -1 and cos_sim(V_x,V_y) = -1 and take their absolute value, isn't the final product still |-1| * |-1| = 1?
> >
> > Based on this major oversight of method design from the authors, I think the submission is not ready for publication yet, so I will maintain my rating.

---

> > > ### Author Response · Authors · 2024-12-01
> > >
> > > Thank you again for your contribution to the review process. With the clock ticking down on the discussion period, we kindly ask if you could revisit our submission, review our rebuttal, and share any additional remarks or conclusions you may have. Your continued engagement will ensure a more thorough and fair assessment of our work, which we believe holds significant potential contributions to the field.

---

> ### Author Response · Authors · 2024-11-26
> **Response to Reviewer h765.**
>
> Thanks for your feedback. Here we conduct the statistics and performance test again.
>
> **Statistics.** We have counted this minor case (`key_sim < 0 and value_sim < 0 and key_sim * value_sim > tau`) on four LLMs using Longbench samples and calculated its percentage among the cases we actually use (`key_sim * value_sim > tau`). The results show that this minor case **scarcely occurs in Llama3 and Mistral**, and it only accounts for **0.25% and 0.02% in Llama2 and LongChat** respectively. Considering that taking this situation into account will instead add a mask operation, it is thus a negligible case.
>
> Test code:
> ```python
> # -- and ++ cases
> mm_pp = torch.count_nonzero(key_sim * value_sim > tau)
> # ++ cases
> pp = torch.count_nonzero((key_sim > 0) * keys_sim * values_sim > tau)
>
> # accumulate all mm_pp and pp, then calculate:
> rate = (mm_pp - pp) / pp
> ```
>
> **Performance test.** We have supplemented the performance using `((key_sim > 0) * (key_sim * v_sim)) > tau`. From the results, it can be seen that **the impact of this minor case on the results is rather minimal and negligible**.
>
> | LLMs          | best baseline | before | after    |
> | ---- | ---- | ---- | ---- |
> | Llama2      | 30.57               | 31.34 | 31.42 (↑) |
> | Llama3      | 37.99               | 38.94 | 38.95 (≈) |
> | LongChat | 14.90               | 31.25 | 31.06 (↓) |
> | Mistral       | 33.86                | 33.93 | 33.92 (≈) |

---

### Official Review · Reviewer_dVvh · 2024-11-03

**Soundness:** 3
**Presentation:** 3
**Contribution:** 2
**Rating:** 5
**Confidence:** 4

**Summary:**

The paper explores efficient KV cache compression for large language models (LLMs) to handle long sequences. It introduces EMS, an Evict-then-Merge Strategy, which uses a Global-Local score to select important tokens, balancing global and local importance. This approach addresses biases from attention weights and positional encoding. EMS compresses the KV cache by first evicting irrelevant tokens, then merging less important ones into class centers. This head-wise method adapts the compression ratio per head, enhancing storage density while preserving model performance.

**Strengths:**

1) A balanced Global-Local score for token selection, reducing bias.
2) A head-wise Evict-then-Merge strategy based on sparsity and redundancy.
3) Efficient parallel compression using a zero-class center for merging.

**Weaknesses:**

1) Please quantify the computation time for calculating global scores and local scores and runtime comparisons between your method and alternatives for different sequence lengths, and determine if you have explored any approximation techniques to reduce this overhead for ultra-long sequences
2) Please quantify the additional computational costs of real-time merging.
3) More experiments are needed: ablation studies varying the relative weights of global and local scores, and performance comparisons across different types of tasks or model sizes.

**Questions:**

1) What are the benefits of doing mean-alignment for sGlo and sLoc?
2) Line 369-371: what are the specific operations of using and not using position information?
3) It's not clearly explained how to distinguish the three categories of KV cache.

---

> ### Author Response · Authors · 2024-11-22
> **Response to Reviewer dVvh. W1, W2**
>
> Thank you so much for acknowledging this paper and for sharing your concerns about algorithm efficiency.
>
> &nbsp;
>
> # W1
>
> For the calculation of the token importance score, **H2O** only calculates global scores, while **SnapKV** only calculates local scores and performs pooling. **EMS** adopts the Global-Local score, calculates both and combines them.
>
> Here we provide the test results with different test sequence lengths using Llama2-7B-chat. All results are obtained by testing 5 times and then averaging the results. The format of the datas in the table is **prefilling time (ms)/decoding time (ms)**.
>
> It's worth mentioning that SnapKV only performs compression in the prefilling stage, so there is no decoding time in the table.
>
> | Method\Seq len                  | 512               | 1024             | 2048            | 3072             | 4096            | Performance |
> | ---- | ---- | ---- | ---- | ---- | ---- | ---- |
> | H2O (global score)            | 0.15/0.08 | 0.14/0.08 | 0.32/0.08 | 0.59/0.08 | 0.96/0.09 | 30.23 |
> | SnapKV (local score)         | 0.26/-         | 0.21/-          | 0.26/-          | 0.37/-          | 0.46/-          | 30.57 |
> | EMS (global-local score) | 0.36/0.26 | 0.36/0.27 | 0.53/0.27 | 0.77/0.28 | 1.14/0.26 | 31.34 |
>
> From the results, the calculation time of the Global-Local score is **basically the sum of the calculation time of the global score and the local score, or slightly less than that**. In addition, the longer the sequence length is, the smaller the additional overhead will be.
>
> Regarding the overhead for calculating the score, EMS focuses on extreme compression while maintaining the performance. We have noticed that some recent works can be adapted to our work to accelerate the compuation .ZipCache [1] utilizes 10% probe tokens to approximate the accumulated scores. Besides, Nacl [2] integrates the calculation of the global score into the FlashAttention kernel. These works are all orthogonal to our work and can be applied to EMS to accelerate the calculation.
>
> [1] ZipCache: Accurate and Efficient KV Cache Quantization with Salient Token Identification, NeurIPS’24, https://arxiv.org/abs/2405.14256.
>
> [2] Nacl: A general and effective kv cache eviction framework for llms at inference time, ACL’24, https://arxiv.org/abs/2408.03675.
>
> &nbsp;
>
> # W2
>
> We set the input length to 4096 and test the merging time under different cache budgets. The experiments are conducted on two RTX4090 GPUs.
>
> | Time\Cache budget | 128   | 256  | 512  | 768  | 1024 |
> | ---- | ---- | ---- | ---- | ---- | ---- |
> | Merging time (ms)    | 1.53 | 1.60 | 1.95 | 2.23 | 2.49  |
> | Total attention time | 3.66 | 3.72 | 3.88 | 4.33 | 4.50  |
>
> Since the code is currently implemented in the initial version, we will conduct further optimizations to enhance the compression speed. Although the merging process takes up a certain amount of inference time, it reduces the memory overhead and also brings a significant performance gain. Here we supplement the performance of different models using more budgets and their comparisons to baselines.
>
> | Method\Cache budget | 128     | 256      | 512     | 768      | 1024  |
> | ---- | ---- | ---- | ---- | ---- | ---- |
> | StreamingLLM                 | 24.39 | 26.99 | 28.92 | 29.46 | 29.77 |
> | H2O                                       | 19.00 | 30.23 | 31.14 | 31.54 | 31.91 |
> | SnapKV                                | 27.78 | 30.57 | 31.64 | 32.05 | 32.27 |
> | EMS                                       | **29.41** | **31.34** | **32.26** | **32.71** | **33.00** |
>
> More results can be seen in **Table 9**. Through the results, we can see that EMS outperforms other methods on the 4 LLMs across different cache budgets. Under an extremely low cache budget, while other baselines may break down, EMS can maintain its performance, which illustrates the robustness of our method.

---

> ### Author Response · Authors · 2024-11-22
> **Response to Reviewer dVvh. W3**
>
> Thanks for pointing out that the relative weight can affect the global-local score. In this paper, we only focus on the impact of global score, local score and global-local score. After some preliminary experiments, we found that changing the relative weight has the potential to achieve better results. This further illustrates the potential of the global-local score as a token importance indicator.
>
> | relative weight | 0(loc)  | 0.1       | 0.25   | 0.33     | 0.5       | 1(glo-loc) | 2           | 3           | 4            | 5            | 10         | Inf(glo) |
> | ---- | ---- | ---- | ---- | ---- | ---- | ---- | ---- | ---- | ---- | ---- | ---- | ---- |
> | Llama2                | 31.06 | 31.02 | 31.00 | 31.08 | 31.27 | 31.34        | **31.49** | 31.37 | 31.19 | 31.15 | 30.71 | 28.81    |
> | Llama3                | 38.52 | 38.54 | 38.71 | 38.84 | 39.11 | 38.94        | 38.89 | 38.93 | 38.68 | 38.58 | 37.99 | 36.33 |
> | LongChat            | 29.32 | 29.56 | 29.94 | 30.28 | 30.45 | **31.25**        | 31.22 | 30.87 | 30.89  | 30.72 | 29.43 | 26.15 |
> | Mistral                  | 33.48 | 33.92 | 33.93 | 33.93 | 33.98 | 33.93        | 33.94 | **34.00** |  33.97 | 33.93 | 33.96 | 29.88 |
>
> **Generally, EMS can also achieve good performance when the relative weight is 1**. We will take the dynamic weights setting as a direction for more exploration in the future. Thanks again for the enhancement of the token importance indicator!

---

> ### Author Response · Authors · 2024-11-22
> **Response to Reviewer dVvh. Q1, Q2, Q3**
>
> # Q1
>
> **Combine global and local scores for less biased indicator.** We observe the intrinsic former and biases of $s_{Glo}$ and $s_{Loc}$, and hope to design a method that can preserve the advantages of both.
>
> **Mean-alignment to counteract the huge scale difference**. Because $s_{Glo}$ accumulates all rows of attention weights, while $s_{Loc}$ only accumulates $L_{win}$ rows. For example, when the sequence length and local window length are 4096 and 32, we did a statistics on the mean values of $s_{Glo}$ and $s_{Loc}$ and they  can differ by a hundredfold.
>
> **Benefits of mean-alignment.** As mentioned in W3, when the relative value of one score is too large, it will lead to performance degradation.
>
> &nbsp;
>
> # Q2
>
> The difference lies in two aspects: the **cosine similarity** and the **position encoding** differences.
>
> **Merge with position information:** with the position information, the cosine similarity between keys decays with the increase of relative distance, leading to **less redundancy and merged chances**. The advantage is that the precise position information of tokens is not changed, which is **beneficial for tasks that require precise token localization**.
>
> **Merge without position information:** redo the positional encoding for each decoding as illustrated in [1]. Thus we can calculate cosine similarity of raw keys (without positional information), and **more tokens can be merged** potentially. Besides, re-apply the positional encoding for each decode is actually superimposing the encoding from 0 to the kv_len involved in the computation, which also **enables streaming generation.**
>
> [1] Efficient Streaming Language Models with Attention Sinks, ICLR’24, https://arxiv.org/abs/2309.17453.
>
> &nbsp;
>
> # Q3
>
> Sorry for the unclarity. Let me illustrate again.
>
> We can first understand it from the **inference** stage. What we actually need to retain are $N_{imp}$ (also known as $N_{budger}$) tokens as class centers. They will be expanded into $\gamma N_{budget}$ tokens according to a look-up-table, and only these tokens will be used for computation.
>
> From the perspective of **compression**, our goal is to compress the most important $\gamma N_{imp}$ tokens into $N_{imp}$ tokens, and evict all the remaining unimportant tokens. In this way, the three types of Key-Value (KV) cache is obvious.  The **first** type consists of $N_{imp}$ compressed class centers. The **second** type includes  $N_{tbm}$(equivalent to $(\gamma-1) N_{imp}$ ) tokens that are to be merged. The **third** type is the evicted irrelevant tokens.

---

> ### Author Response · Authors · 2024-11-24
>
> Thank you for your valuable feedback on our submission. We greatly appreciate your efforts and constructive comments. As a kind reminder, the discussion period is drawing close. Please let us know if there remains anything that we can further clarify to improve our work. We understand that this is a busy period and appreciate your time and consideration.

---

> ### Author Response · Authors · 2024-12-01
>
> Thank you again for your contribution to the review process. With the clock ticking down on the discussion period, we kindly ask if you could revisit our submission, review our rebuttal, and share any additional remarks or conclusions you may have. Your continued engagement will ensure a more thorough and fair assessment of our work, which we believe holds significant potential contributions to the field.

---

### Official Review · Reviewer_U976 · 2024-11-03

**Soundness:** 3
**Presentation:** 2
**Contribution:** 3
**Rating:** 5
**Confidence:** 4

**Summary:**

This paper introduces an adaptive Evict-Then-Merge Strategy (EMS) for efficient key-value (KV) cache compression. EMS utilizes a combined global-local scoring mechanism that tracks attention accumulation both across the entire model (global) and within individual layers (local) to rank and identify essential tokens. To meet the target compression budget, EMS first evicts tokens deemed irrelevant based on these criteria. It then merges the remaining tokens based on similarity; if tokens are too dissimilar, they are merged with a zero token. Empirical results with a budget of 256 demonstrate EMS’s superior performance in LongBench compared to other state-of-the-art (SoTA) methods.

**Strengths:**

The paper proposed EMS, which incorporates global and local attention to determine the token's importance for the KV cache, is a good idea. The strategy used to evict and merge tokens is unique, and under the proposed budget of 256 tokens, EMS performs well in LongBench.

**Weaknesses:**

I have several comments and questions that I hope the authors can clarify:

1. Memory Management: Could the authors confirm my understanding? It appears that EMS operates on a per-head basis, with each head tracking its own $s_{GLo}$ and $s_{Loc}$ statistics. If this is correct, then values such as $N_{tbm}$, $N_{irr}$, and $N_{imp}$ are non-uniform across all heads, meaning the KV cache would need to exceed the target budget to support this irregular pattern and maintain data contiguity. How was this challenge addressed?

2. Throughput: Due to the irregular pattern, EMS employs “shared entry expansion” to enable matrix vectorization. However, there is a non-zero probability that shared entries could expand to the full context length, yet this potential has not been discussed. Could the authors elaborate on how they handle or mitigate this?

3. Efficiency Analysis: Given that EMS is dynamic, there’s a non-zero probability of memory spikes due to irregular patterns, particularly in scenarios where few tokens overlap across heads. This variability could impact efficiency, making the analysis in Section 5.4 potentially misleading. Could the authors clarify how they accounted for this in their analysis?

4. Budget Restriction: I am curious about the decision to restrict the budget to 256 in LongBench. How was this value determined? Typically, we set the budget to the minimum level that maintains performance parity with a full cache, yet I didn’t find a discussion on the rationale behind this choice.

**Questions:**

see weaknesses

---

> ### Author Response · Authors · 2024-11-22
> **Response to Reviewer U976. W1**
>
> Thank you very much for recognizing this paper and for raising concerns about memory.
> We believe there may be a slight misunderstanding regarding our method.
>
> > values such as $N_{tbm}$, $N_{irr}$ and $N_{imp}$ are non-uniform across all heads
> >
>
> **Uniform head budgets**: EMS does track $s_{Glo}$ and  $s_{Loc}$  for each head. But **$N_{tbm}$, $N_{irr}$ and $N_{imp}$ are same for each head**, and the length of $s_{Glo}$ and $s_{Loc}$ are also the same and constant $\gamma N_{imp}$. So there won’t exist non-uniform problem.
>
> > support this irregular pattern and maintain data contiguity
> >
>
> **Adapt irregular patterns by unified merging**: we speculate that the irregular pattern you are concerned about lies in the possibility of different merge ratios in the unified
>
> > the KV cache would need to exceed the target budget
> >
>
> **Scalarized score indicator is much smaller than vectorized KV tokens:** for example, the $\gamma=4$ (ablated in Table 7), head_dim=128, the additional overhead of score indidator accounts for $(4*3*N_{imp})/(N_{imp}*128*2)=4.7%$ of the retained KV cache, which is efficient.

---

> ### Author Response · Authors · 2024-11-22
> **Response to Reviewer U976. W2**
>
> Thanks for your comment.
>
> Irregular pattern illustration: EMS set the same KV cache budget and expansion length for each head. The head-wise characteristic is utilized by expanding to the evicted zero or the merged values, which means that different heads have different expanded patterns.
>
> **The expansion length is indicated by the factor $\gamma$.** EMS will first evict $N_{irr}$ tokens, and then perform a merge within the remaining $N_{imp}+N_{tbm}=\gamma N_{imp}$ tokens. The compressed $N_{imp}$ tokens are expanded to $\gamma N_{imp}$ tokens during the inference.
>
> In Table 7, we have tested the impact of $\gamma$. Setting $\gamma=$'Inf' is equivalent to expanding shared entries to the full context length. And we found that expanding to full context length is sub-optimal, which might be due to the distraction of long context. So the the eviction in the first stage can not only reduce overheads but also improve performance.
>
> | $\gamma$ | 1           | 2            | 3            | 4           | 5            | 6           | 7            | 8            | Inf        |
> |  ----  | ----  | ----  | ----  | ----  | ----  | ----  | ----  | ----  | ----  |
> | Avg.                | 30.05 | 30.88 | 31.00 | 31.34 | 31.34 | 31.25 | 31.04 | 30.92 | 31.18 |

---

> ### Author Response · Authors · 2024-11-22
> **Response to Reviewer U976. W3**
>
> Thanks for your comment.
>
> We believe that the expansion will not cause this problem. The compressed tokens themselves exist in a continuous memory space. The way we obtain the expanded tokens is to first allocate a continuous memory space and then expand the compressed tokens according to the position look-up-table(LUT). The overhead brought by expansion is only for the currently computed layer, and other layers will still maintain compact storage. In fact, this is a trade-off among performance, memory, and latency. Although there is additional overhead, it leads to more compact memory and better performance.

---

> ### Author Response · Authors · 2024-11-22
> **Response to Reviewer U976. W4**
>
> Thanks for your comment.
>
> We set the cache budget 256 to show that **even at very low budgets, EMS holds up well**. In fact, the maximum context lengths for Llama-2-7B-Chat, Llama-3-8B-Instruct, LongChat-7B-v1.5-32k and Mistral-7B-Instruct-v0.2 are 4096, 8192, 32k and 32k. So the budget 256 is a very small budget and accounts for **6.25%, 3.125%, 0.8% and 0.8%** for these four models. Even under this circumstance, our method can maintain **95.7%, 93.7%, 92.7%, 85.5%** performance  with a full cache.
>
> To further validate the performance of EMS, we also tested the performance of different models using more budgets and their comparisons to baselines.
>
> Here we show the performance on Llama2 using different budgets and the comparisons with baselines. More results can be seen in **Table 9**.
>
> | Method                 | 128     | 256      | 512     | 768      | 1024  |
> | ---- | ---- | ---- | ---- | ---- | ---- |
> | StreamingLLM | 24.39 | 26.99 | 28.92 | 29.46 | 29.77 |
> | H2O                       | 19.00 | 30.23 | 31.14 | 31.54 | 31.91 |
> | SnapKV                 | 27.78 | 30.57 | 31.64 | 32.05 | 32.27 |
> | EMS                        | **29.41** | **31.34** | **32.26** | **32.71** | **33.00** |
>
> Through the results, we can see that EMS outperforms other methods on the 4 LLMs across different cache budgets.

---

> ### Author Response · Authors · 2024-11-24
>
> Thank you for your valuable feedback on our submission. We greatly appreciate your efforts and constructive comments. As a kind reminder, the discussion period is drawing close. Please let us know if there remains anything that we can further clarify to improve our work. We understand that this is a busy period and appreciate your time and consideration.

---

> > ### Comment · Reviewer_U976 · 2024-11-25
> > **Response to Authors**
> >
> > I thank the authors for their detailed responses. After reviewing other reviewers' concerns, I found that several key challenges have not yet been adequately addressed. The discrepancy in cosine similarity is particularly noteworthy, as pointed out by Reviewer h765. Therefore, I will maintain my score. I am open to raising the score once this is addressed.

---

> > > ### Author Response · Authors · 2024-11-26
> > > **Response to Reviewer U976.**
> > >
> > > Thanks for your attention. We have addressed the concerning raised by reviewer h765.
> > >
> > > We conduct the statistic and performance test again.
> > >
> > > **Statistics.** We have counted this minor case (`key_sim < 0 and value_sim < 0 and key_sim * value_sim > tau`) on four LLMs using Longbench samples and calculated its percentage among the cases we actually use (`key_sim * value_sim > tau`). The results show that this minor case **scarcely occurs in Llama3 and Mistral**, and it only accounts for **0.25% and 0.02% in Llama2 and LongChat** respectively. Considering that taking this situation into account will instead add a mask operation, it is thus a negligible case.
> > >
> > > Test code:
> > >
> > > ```python
> > > # -- and ++ cases
> > > mm_pp = torch.count_nonzero(key_sim * value_sim > tau)
> > > # ++ cases
> > > pp = torch.count_nonzero((key_sim > 0) * keys_sim * values_sim > tau)
> > >
> > > # accumulate all mm_pp and pp, then calculate:
> > > rate = (mm_pp - pp) / pp
> > > ```
> > >
> > > **Performance test.** We have supplemented the performance using `((key_sim > 0) * (key_sim * v_sim)) > tau`. From the results, it can be seen that **the impact of this minor case on the results is rather minimal and negligible**.
> > >
> > > | LLMs          | best baseline | before | after    |
> > > | ---- | ---- | ---- | ---- |
> > > | Llama2      | 30.57               | 31.34 | 31.42 (↑) |
> > > | Llama3      | 37.99               | 38.94 | 38.95 (≈) |
> > > | LongChat | 14.90               | 31.25 | 31.06 (↓) |
> > > | Mistral       | 33.86                | 33.93 | 33.92 (≈) |

---

> > > ### Author Response · Authors · 2024-12-01
> > >
> > > Thank you again for your contribution to the review process. With the clock ticking down on the discussion period, we kindly ask if you could revisit our submission, review our rebuttal, and share any additional remarks or conclusions you may have. Your continued engagement will ensure a more thorough and fair assessment of our work, which we believe holds significant potential contributions to the field.

---

### Meta-Review · Area_Chair_VjFi · 2024-12-21

**Metareview:**

The paper introduces EMS, a novel strategy for KV cache compression in large language models. It employs a combined Global-Local scoring mechanism to rank token importance and an adaptive Evict-then-Merge strategy to compress cache, achieving impressive performance under extreme compression ratios. The authors demonstrate EMS's efficacy through metrics such as perplexity and retrieval accuracy on various benchmarks.

The strengths of the submission lie in its creative use of global-local scoring to balance attention biases and its innovative compression framework that efficiently manages token sparsity across heads. The results, which showcase EMS's competitive performance, are promising and indicate potential for practical application in resource-constrained scenarios.

However, the submission has notable weaknesses. The evaluation is limited to a fixed cache budget, which restricts the robustness of conclusions. Concerns regarding computational overhead, particularly for the global scoring mechanism, remain unresolved, making deployment infeasible for some real-world use cases. Additionally, discrepancies in experimental baselines and the lack of clarity in some methodological details, such as redundancy estimation and expansion mechanisms, weaken the overall impact of the work.

While the paper demonstrates innovative contributions, these limitations, particularly in computational feasibility and the narrow evaluation scope, lead us to recommend rejection at this stage. Addressing these issues and providing more comprehensive experiments could significantly strengthen the work.

**Additional Comments On Reviewer Discussion:**

The reviewers raised valid concerns about computational efficiency and methodological clarity, which the authors partially addressed during the discussion. However, key issues, such as the practicality of the global scoring mechanism and experimental discrepancies, were not convincingly resolved. Engaging with these critiques in future iterations would improve the submission's quality and impact.

---

### Decision · Program_Chairs · 2025-01-22

Reject